# CD31 signaling promotes the detachment at the uropod of extravasating neutrophils allowing their migration to sites of inflammation

Francesco Andreata[1†], Marc Clément[1†], Robert A Benson[2], Juliette Hadchouel[3], Emanuele Procopio[1], Guillaume Even[1], Julie Vorbe[1], Samira Benadda[4], Véronique Ollivier[1], Benoit Ho-Tin-Noe[1], Marie Le Borgne[1], Pasquale Maffia[2,5], Antonino Nicoletti[1], Giuseppina Caligiuri[1,6]*

[1]Université Paris Cité and Université Sorbonne Paris Nord, INSERM, Laboratory for Vascular Translational Science (LVTS), Paris, France; [2]Centre for Immunobiology, Institute of Infection, Immunity and Inflammation, College of Medical, Veterinary and Life Sciences, University of Glasgow, Glasgow, United Kingdom; [3]Université Paris Cité, INSERM, Paris Cardiovascular Research Center (PARCC), Paris, France; [4]Cell and Tissue Imaging Platform, INSERM, CNRS, ERL8252, Centre de Recherche sur l'Inflammation (CRI), Paris, France; [5]Department of Pharmacy, School of Medicine and Surgery, University of Naples Federico II, Naples, Italy; [6]Department of Cardiology and of Physiology, Assistance Publique-Hôpitaux de Paris (AP-HP), Hôpitaux Universitaires Paris Nord Val-de-Seine, Site Bichat, Paris, France

*For correspondence:
giuseppina.caligiuri@inserm.fr

†These authors contributed equally to this work

**Abstract** Effective neutrophil migration to sites of inflammation is crucial for host immunity. A coordinated cascade of steps allows intravascular leukocytes to counteract the shear stress, transmigrate through the endothelial layer, and move toward the extravascular, static environment. Those events are tightly orchestrated by integrins, but, while the molecular mechanisms leading to their activation have been characterized, the regulatory pathways promoting their detachment remain elusive. In light of this, it has long been known that platelet-endothelial cell adhesion molecule (*Pecam1*, also known as CD31) deficiency blocks leukocyte transmigration at the level of the outer vessel wall, yet the associated cellular defects are controversial. In this study, we combined an unbiased proteomic study with in vitro and in vivo single-cell tracking in mice to study the dynamics and role of CD31 during neutrophil migration. We found that CD31 localizes to the uropod of migrating neutrophils along with closed $\beta_2$-integrin and is required for essential neutrophil actin/integrin polarization. Accordingly, the uropod of *Pecam1*[-/-] neutrophils is unable to detach from the extracellular matrix, while antagonizing integrin binding to extracellular matrix components rescues this in vivo migratory defect. Conversely, we showed that sustaining CD31 co-signaling actively favors uropod detachment and effective migration of extravasated neutrophils to sites of inflammation in vivo. Altogether, our results suggest that CD31 acts as a molecular rheostat controlling integrin-mediated adhesion at the uropod of egressed neutrophils, thereby triggering their detachment from the outer vessel wall to reach the inflammatory sites.

## Editor's evaluation

The demonstration of the hemophilic adhesion molecule CD31's localization to the migrating neutrophil's uropod and required signaling through its ITIM domain are valuable findings.

Convincing evidence now includes defective extravasation of CD31 ITIM motif-deleted neutrophils in adoptive transfer experiments. Based on the critical role of leukocyte transmigration in many physiologic and pathologic processes, this work will be of interest to a broad audience.

## Introduction

One of the hallmarks of a successful immune response is the capacity of leukocytes to reach the sites of pathogenic stimuli. This notion is particularly true for neutrophils, the most abundant leukocyte population in the human blood, and crucial innate immune effectors (*Ley et al., 2018*). Mature neutrophils incessantly patrol and scan microvascular endothelial cells (ECs), searching for extravasation cues locally driven by chemoattractants and adhesion molecules present at the surface of post-capillary venules. When appropriate, a spatiotemporally coordinated adhesion cascade on inflamed ECs allows neutrophils to counteract vascular shear stress and extravasate through the endothelial cell layer (*Schnoor et al., 2021*). Effective migration toward the inflammatory site occurs only after the final detachment from the outside vascular structure (*Hyun et al., 2012*). This is a fundamental but poorly characterized process that requires a quick amendment of the highly adhesive phenotype established to counteract flow conditions within the bloodstream (*Pouwels et al., 2013*) in order to migrate through the extravascular static environment.

Because neutrophils are endowed with harmful cytotoxic compounds, their aberrant recruitment and retention at inflammatory sites might result in severe bystander injury to the tissue (*Laforge et al., 2020*). Therefore, deciphering the molecular mechanisms of neutrophil trafficking is of paramount importance for identifying new therapeutic targets that control excessive inflammatory responses without impairing host defense.

Under dynamic blood conditions, neutrophils rapidly establish adhesive connections with the endothelial selectins. These engagements induce an immunoreceptor tyrosine-based activation motif (ITAM)-dependent conformational change of $\beta_2$-integrins (LFA-1 and Mac-1), which increases their affinity configuration and allows neutrophil arrest within the vessel (*Abram and Lowell, 2007*; *Zarbock et al., 2008*). Here, neutrophils must hold integrins in their maximal adhesive state to prevent them from being flushed away by the hydrodynamic forces of the bloodstream (*Thome et al., 2018*). Once adhering to ECs, their shape changes noticeably from an almost spherical to a flattened shape due to the relocalization of the actin cytoskeleton and cell adhesion networks. After breaching the vessel wall, neutrophils interact with the basement vascular matrix constituents, searching for less dense, permissive regions for their passage (*Wang et al., 2006*). At the rear, the uropod of migratory neutrophils is characterized by a highly contractile and low adhesive state, which is important for ensuring efficient migration through inflammatory sites (*Hind et al., 2016*; *Pouwels et al., 2013*). However, intravital observation of extravasating neutrophils has shown that, while their leading edge moves forward within the interstitial tissue, the uropod initially remains 'trapped' at the level of the basolateral part of the EC junctions until integrin closure allows their detachment (*Hind et al., 2016*; *Wang et al., 2006*). Although this step is emerging as key for proper leukocyte recruitment, the underlying molecular mechanisms remain poorly defined.

When blocking platelet-endothelial cell adhesion molecule (*Pecam1*, also known as CD31), most extravasating neutrophils fail to reach inflammatory sites and remain trapped on the basolateral side out of the microvessels (*Wakelin et al., 1996*). This finding was also recapitulated in *Pecam1^{-/-}* mice (*Duncan et al., 1999*), yet more than 30 y later the function of CD31 in neutrophils is not completely understood.

CD31 is an immunoglobulin (Ig)-like protein known to act as a co-inhibitory receptor on lymphocytes due to the presence of two immunoreceptor tyrosine-based inhibitory motifs (ITIMs) in its cytoplasmic tail. Upon phosphorylation, CD31 ITIMs become docking sites for Src Homology 2 (SH2)-containing phosphatases, uncoupling ITAM-dependent leukocyte activation (*Henshall et al., 2001*; *Marelli-Berg et al., 2013*). Being expressed by all hematopoietic cells and ECs, CD31 ITIM signaling properties confer a pleiotropic and central role to CD31 in maintaining homeostasis at the blood–vessel interface (*Caligiuri, 2020*). The fact that extravasating neutrophils can engage in CD31:CD31 trans-homophilic interactions when crossing endothelial lateral cell junctions – where CD31 is particularly enriched – suggests a role for CD31 in the migration cascade. Since integrin opening and outside-in signaling are

also ITAM-dependent (*Bezman and Koretzky, 2007*), we investigated whether ITIM-associated CD31 signaling could be involved in fine-tuning adhesion during the extravasation process.

Using an unbiased proteomic-based approach and functional experiments, combining in vitro and in vivo live microscopy, we uncovered the role of CD31 signaling in favoring integrin closure at the uropod of extravasating neutrophils, thus allowing their detachment from the basement membrane and efficient migration toward sites of inflammation.

## Results

### Neutrophil recruitment is accompanied by CD31 signaling, while its absence compromises cell detachment from the extravascular component of the vessel

We used an acute peritonitis model induced by intraperitoneal (i.p.) injection of IL-1β and quantified neutrophil migration within the extravascular space in WT and *Pecam1*[-/-] mice by flow cytometry. In WT mice, neutrophils started to migrate and accumulate in the peritoneal cavity within 4 hr after IL-1β injection (*Figure 1A*). However, in *Pecam1*[-/-] mice, neutrophils started to migrate to the peritoneal cavity during the first 2 hr, but eventually failed to extravasate within the peritoneal fluid at later time points (*Figure 1A*). This difference between WT and *Pecam1*[-/-] mice was not accompanied by a difference of neutrophil number increase in the bloodstream (*Figure 1B*), nor was there a defective molecular response to IL-1β in terms of chemokines involved in neutrophil mobilization from the bone marrow (SDF-1), local endothelial/platelet activation at sites of extravasation (P-selectin), specific chemoattraction at the inflammatory site (CXCL1), or global acute-phase reaction (PTX3) (*Figure 1C*). Next, we analyzed the spatial distribution of neutrophils by fluorescence microscopy through the microvessel wall in the omentum, an important site of neutrophil extravasation in the peritoneum (*Buscher et al., 2016*). Neutrophils from WT mice were located within the extravascular space (*Figure 1D*). Notably, the phosphorylation level of CD31 (pY713) was inversely correlated with neutrophil distance from the vessel wall (i.e., increased pY713 when cells were close to the blood vessel wall), suggesting that CD31 signaling in neutrophils was active during extravasation (*Figure 1E*). Although most neutrophils from *Pecam1*[-/-] mice were also found to extravasate (*Figure 1F*, inset 3D section), their migration was delayed as they remained in close contact with the outer layer of the vessel wall (*Figure 1F–H*). Taken together, these results indicate that CD31 is involved in facilitating the detachment of neutrophils from the outer components of the vessel wall after extravasation. This detachment is crucial for the timely migration to the inflammatory site.

### CD31 associates with integrin and actin networks in human neutrophils

To gain further mechanistic insight into the molecular networks in which CD31 is involved, we performed proteomic analysis to identify the CD31 interactome. Proteins associated with CD31 were studied by unbiased proteomic analysis of human neutrophils stimulated with N-formylmethionine-leucyl-phenylalanine (fMLP, a synthetic peptide commonly used as a chemoattractant to induce cell migration and activation of neutrophils) compared to neutrophils maintained under resting conditions. Immunoprecipitated CD31 molecules from cell lysates were subjected to Orbitrap to identify CD31-associated partners (*Figure 2A*). We identified 173 protein co-immunoprecipitated with CD31 molecules (*Figure 2B*, *Supplementary file 1*). A total of 110 proteins were present in both resting and fMLP-stimulated conditions, whereas 43 proteins were only associated with CD31 molecules under resting conditions and 20 proteins were specifically detected under fMLP stimulation (*Figure 2B*, *Supplementary file 1*). Among these, we found known CD31-interacting proteins in chemokine-stimulated neutrophils, such as GRB2, CD177, SHP, and MSN (*Sachs et al., 2007*), as well as ITIM-interacting SH2 phosphatase SHIP1 (*Pumphrey et al., 1999*), and integrins such as CD11b and CD18 (*Figure 2C*, *Supplementary file 1*). Ingenuity pathway analysis (IPA) and KEGG functional annotation pathways revealed that the CD31 interactome was linked to the integrin and actin networks, under resting and activated conditions, consistent with the role of CD31 in neutrophil migration (*Figure 2D* and *Figure 2—figure supplement 1A and B*). Based on the IPA results, we elaborated a model of protein interaction showing how CD31, through SHIP1 (INDPP5), regulates cytoskeleton reorganization and integrin activation during neutrophil migration (*Figure 2E*).

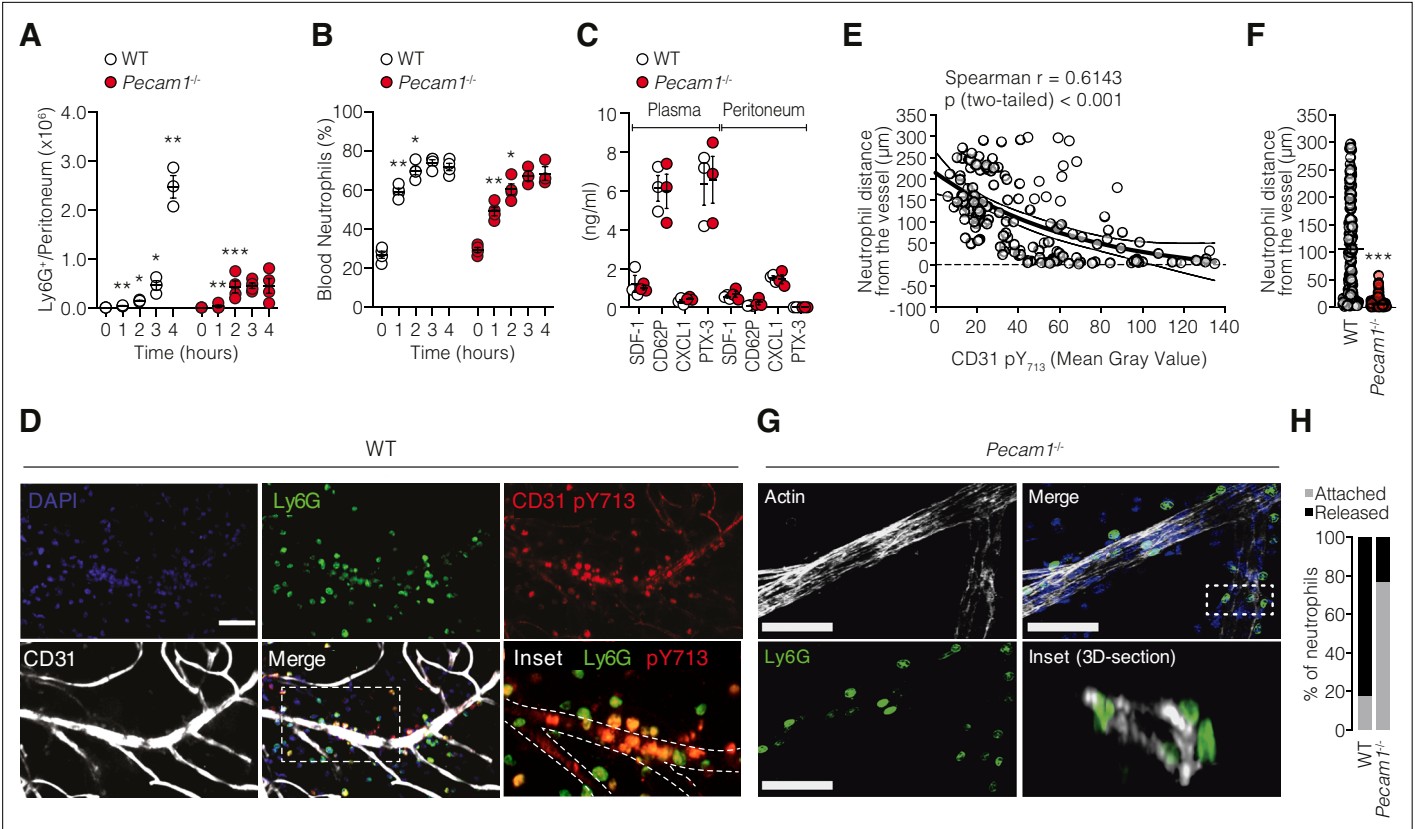

**Figure 1.** *Pecam1*[-/-] neutrophils exhibit efficient mobilization in the circulation but fail to reach the inflammatory site after crossing the vessel wall. (**A**) Absolute number of Ly6G[+] cells harvested in the peritoneal cavity at different time points after the induction of sterile peritonitis in WT or *Pecam1*[-/-] mice. (**B**) Blood neutrophil levels at the indicated time points. (**C**) Concentration of SDF-1, CD62P, CXCL1, and PTX-3 in the plasma and cell-free peritoneal fluid of *Pecam1*[-/-] and WT mice at 4 hr after PBS or IL-1β injection. Data are presented as mean ± SE. n = 3/5 per condition. The data represent three independent experiments. Statistical significance is indicated relative to the previous time point. Mann–Whitney test: *p<0.05, **p<0.01, ***p<0.001. (**D**) Confocal micrographs of whole-mount omentum showing extravasating Ly6G[+] cells (clone 1A8; green) from a post-capillary venule with CD31 expression (polyclonal, R&D #AF3628; white). Samples were also stained with a monoclonal antibody against phosphorylated CD31 tyrosine 713 (pY713, clone EPR8079; red staining). Inset shows colocalization of pY713 and Ly6G. Scale bar 50 μm. (**E**) Inverse correlation between pY713 staining intensity of each Ly6G[+] cell and the distance from the vessel (Spearman correlation and exponential one-phase decay regression with 95% confidence interval). (**F**) Quantification of neutrophil distance from the closest vessel in WT and *Pecam1*[-/-] mice 4 hr after IL-1β injection. n = 153 and n = 89 cells for WT and *Pecam1*[-/-] mice, respectively. Unpaired nonparametric Mann–Whitney test: ***p<0.001. (**G**) Ly6G[+] cells (green) accumulate around the outer edge of post-capillary venules (phalloidin staining, white) in the omentum of *Pecam1*[-/-] mice. Scale bar 50 μm. (**H**) Percentage of neutrophils attached to the outer part of the vessels (gray) and percentage of neutrophils released into the extravascular space (black) in WT and *Pecam1*[-/-] mice.

The online version of this article includes the following source data for figure 1:

**Source data 1.** Quantification of total number of neutrophils in WT and *Pecam1*[-/-] mice after IL-1β challenging (*Figure 1A*).

**Source data 2.** Fraction of neutrophils in the peripheral blood of WT and *Pecam1*[-/-] mice after IL-1β challenging (*Figure 1B*).

**Source data 3.** Quantification of inflammatory cytokines in the plasma or peritoneal fluid of WT and *Pecam1*[-/-] mice 4 hr after IL-1β challenging (*Figure 1C*).

**Source data 4.** Spearman correlation between the distance of each neutrophil from the closest capillary venule and its CD31 phospho(p)-Tyrosine(Y)–713 level (*Figure 1E*).

**Source data 5.** Distance (μm) of neutrophils from the closest capillary venule in the omentum of WT and *Pecam1*[-/-] mice upon IL-1β challenge (n = 3/group) (*Figure 1F*).

**Source data 6.** Fraction of neutrophils attached (<5 μm) or released (>5 μm) from the closest capillary venule in the omentum of WT and *Pecam1*[-/-] mice upon IL-1β challenge (n = 3/group) (*Figure 1H*).

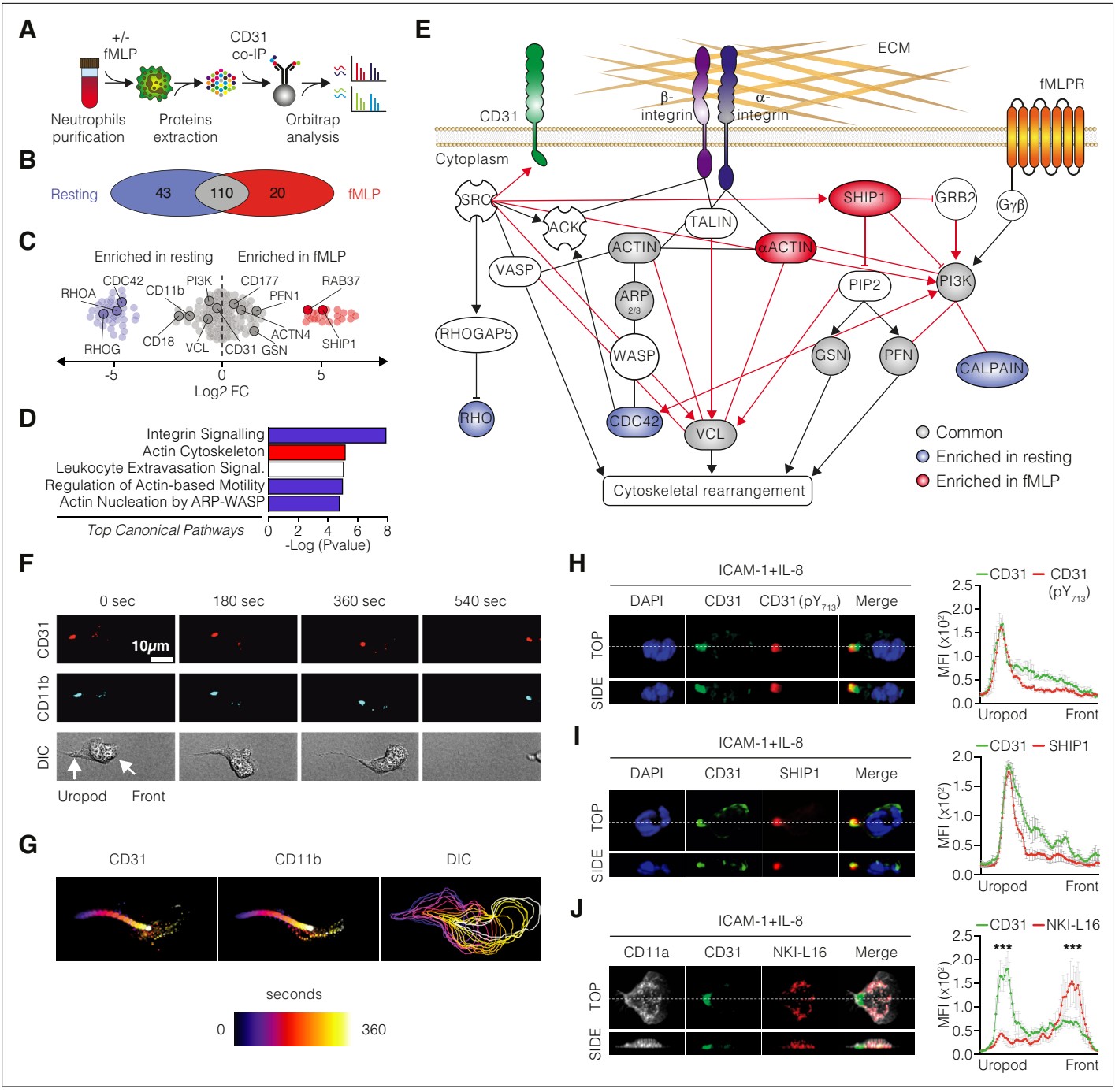

**Figure 2.** Dynamic involvement of CD31 in integrin molecular complexes at the neutrophil uropod. (**A**) Experimental strategy of shotgun proteomic analysis used to identify CD31 partners in human neutrophils under resting or fMLP-activated conditions. (**B–D**) Venn diagrams (**B**) and dot plots (**C**) illustrating the proteins identified by mass spectrometry upon CD31 co-immunoprecipitation in resting and fMLP-activated human neutrophils. (**D**) Ingenuity analysis of the top canonical pathways enriched in the CD31 interactome. The x-axis represents the statistical score, while the y-axis represents the corresponding canonical pathways. Blue color indicates negative Z scores (pathway suppression), red color indicates positive Z scores (pathway activation), and white color indicates a neutral Z score. (**E**) Ingenuity network analysis of the CD31 interactome under resting (light blue) or activated conditions (red), with unchanged partners depicted in gray. (**F**) Timeframe images from live video microscopy showing neutrophils stained with fluorophore-coupled monoclonal antibodies against CD11b (light blue) and CD31 (red) moving on a fibronectin-coated surface. (**G**) Cell motion over time is visualized by a time color-coded superposition of frames. (**H**) IL-8-stimulated primary human neutrophils migrating on recombinant human ICAM-1 stained for CD31 (clone WM59; green) and phosphorylated CD31 ITIM tyrosine 713 (pY713, clone EPR8079; red). (**I**) Staining for SHIP-1 (clone P1C1; red) or (**J**) total CD11a (clone TS2/4; white) and CD11a opened/active conformation (clone NKI-L16, red) in IL-8-stimulated primary human neutrophils. Representative 3D reconstructions of top and side views are shown at the bottom of each panel. Quantification of mean fluorescence

*Figure 2 continued on next page*

*Figure 2 continued*

intensities (MFIs) for the indicated staining along the cell axis is also presented (mean ± SE [SE bars in gray color]; n > 10 cells/staining; statistical comparisons between the first 20 frontal and last 20 posterior measurement points). *p<0.05, **p<0.01, ***p<0.001 (Mann–Whitney test).

The online version of this article includes the following source data and figure supplement(s) for figure 2:

**Source data 1.** Protein pulled down with CD31 and identified by mass spectrometry (*Figure 2C*).

**Source data 2.** Quantification of mean fluorescence intensities (MFIs) of CD31 (clone WM59) and CD31 pY713 (clone EPR8079) along the cell axis (*Figure 2H*).

**Source data 3.** Quantification of mean fluorescence intensities (MFIs) of CD31 (clone WM59) and SHIP-1 (clone P1C1) along the cell axis (*Figure 2I*).

**Source data 4.** Quantification of mean fluorescence intensities (MFIs) of CD31 (clone WM59), total CD11a (clone TS2/4), and opened CD11a (clone NKI-L16) along the cell axis (*Figure 2J*).

**Figure supplement 1.** Orbitrap data analysis for the identification of CD31 pathways in human neutrophils.

**Figure supplement 1—source data 1.** KEGG functional annotation pathways enriched in human neutrophil CD31 interactome under resting conditions.

**Figure supplement 1—source data 2.** KEGG functional annotation pathways enriched in human neutrophil CD31 interactome under fMLP-activated conditions.

## Functional CD31 dynamically engages at the uropod of migrating neutrophils

Since Orbitrap analysis suggested the involvement of CD31 in the integrin network, we analyzed the in vitro spatiotemporal dynamics of CD31 using live imaging under resting versus chemotactic conditions. We found that both CD31 and CD11b rapidly relocated to the uropod during neutrophil migration (*Figure 2F and G* and *Video 1*), while a minor fraction of both molecules remained localized at the leading edge. Considering that neutrophil uropod is characterized by a low-adhesion state that facilitates rear detachment during forward cellular movements (*Hind et al., 2016*), we hypothesized that CD31 could have been involved in controlling β2-integrins (CD11b:CD18 and CD11a:CD18 heterodimers) conformation via inside-out signaling. To this end, we explored the functionality of CD31 and the activation state of integrins using multicolor confocal microscopy in human neutrophils interacting with immobilized ICAM-1 – one of the major β2-ligands for migrating leukocytes – and IL-8.

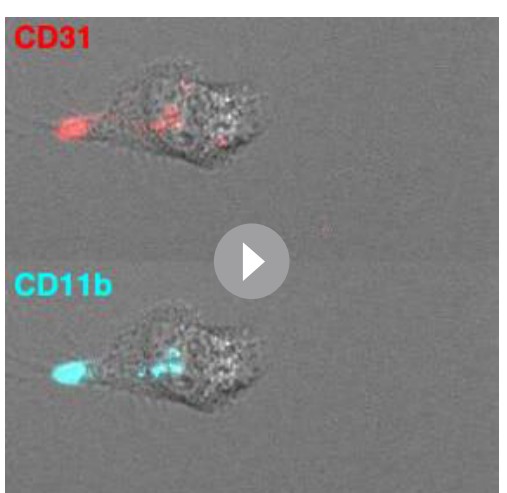

**Video 1.** Live video microscopy of purified human neutrophils stained with anti CD31 and anti CD11b fluorescent antibody moving on fibronectin-coated surface. Timeframe images from live video microscopy of neutrophils stained with fluorophore-coupled monoclonal antibodies directed against CD11b (light blue) and CD31 (red) moving onto a fibronectin-coated surface.

https://elifesciences.org/articles/84752/figures#video1

Under these conditions, we found that CD31 localizes preferentially to the uropod and, only at the rear, CD31 displayed phosphorylated ITIM motifs (*Figure 2H*). In addition, SH2-containing inositol phosphatase SHIP1, which has been found in the CD31 interactome, was also concentrated at the cellular rear (*Figure 2I*), supporting the notion that functional CD31 molecular clusters are present in the uropods of motile neutrophils. This prompted us to perform a detailed analysis of the activation state of integrins localized with CD31. Taking advantage of the conformation-specific antibody NKI-L16, which specifically recognizes an epitope accessible only in the activated form of the integrin alpha L chain CD11a (*Lefort et al., 2012*), we examined the distribution of CD31 molecules in relation to total CD11a molecules (regardless of their conformation) and activated CD11a molecules (NKI-L16-positive). Interestingly, the staining patterns for CD31 and extended β2-integrins were mutually exclusive. Clusters positive for NKI-L16 (indicating active integrins, shown in red) were exclusively localized at the leading edge, where CD31 (shown in green) was absent. Conversely, the staining with this antibody was

absent at the uropod, where most CD31 was concentrated (*Figure 2J*). The co-localization of functional CD31 clusters with integrins that did not react with the NKI-L16 antibody indicates that CD31 is enriched at the uropod along with closed β2-integrins, in contrast to fully opened integrins that are restricted to the leading edge. Taken together, these findings suggest that CD31 signaling may contribute to maintaining integrins in a closed/bent conformation at the uropod, the region where cell detachment is required, during neutrophil migration.

## CD31 genetic deficiency endows neutrophils with aberrant uropod organization

We investigated how CD31 knockout could affect integrin polarization in neutrophils interacting with laminin α4, an important constituent of the basement membrane involved in neutrophil extravasation (*Wang et al., 2006*). Since talin is known to act as a bridge between the cytoskeleton and active integrins (*Calderwood et al., 2013*), we therefore observed the distribution of talin and actin filaments within the cells to evaluate the polarization state of mouse neutrophils. In agreement with previous studies (*Pouwels et al., 2013*), wild-type neutrophils quickly polarized with a leading-edge rich in F-actin, while the integrin-associated protein talin accumulated at the uropod, where the location of the two proteins was almost mutually exclusive (*Figure 3A and B*). The fact that talin was spatially far from F-actin at the cell rear indicated that the integrins at the uropod of WT leukocytes were in a low-adhesive conformation state. Instead, such polarization was clearly defective in *Pecam1⁻/⁻* neutrophils, where F-actin was concentrated along with talin also at the cellular rear (*Figure 3C*) so that *Pecam1⁻/⁻* neutrophils appeared particularly stretched and displayed more elongated uropods compared to wild-type cells (*Figure 3D and E*), thus reflecting an abnormal subcellular organization in the absence of CD31.

Since CD44 localizes at the uropod during cellular migration (*Pouwels et al., 2013*), we stained this molecule to identify the rear of cells moving onto immobilized recombinant mouse laminin α4. Whereas in wild-type neutrophils SHIP1 was preferentially localized together with CD44 at the site of a clearly identifiable uropod (*Figure 3F*), CD44 and SHIP1 were both distributed more evenly throughout the cell body in *Pecam1⁻/⁻* neutrophils. These findings further supported the hypothesis that CD31 was critically involved in uropod formation of neutrophils interacting with laminin.

Finally, to assess whether CD31 signaling was required for triggering neutrophil polarization, we generated mice expressing a mutated version of CD31 by floxing exons 13 and 14, which encode the ITIM intracellular motifs (*Pecam1^ITIM-/-*, *Figure 3—figure supplement 1*). Experiments performed with these mice phenocopied the data obtained using whole *Pecam1⁻/⁻* deficiency (*Figure 3H–K*), demonstrating that CD31 signaling properties are mandatory for proper polarization of migrating neutrophils.

## CD31 signaling unleashes neutrophil recruitment to inflamed sites in vivo

In our previous studies, we demonstrated that P8RI, a CD31 agonist peptide, effectively rescued and sustained the functionality of CD31 ITIM in T lymphocytes (*Clement et al., 2015*; *Fornasa et al., 2012*; *Fornasa et al., 2010*) and macrophages (*Andreata et al., 2018*), thereby providing promising potential for theranostic applications (*Hoang et al., 2018*; *Vigne et al., 2019*). In the context of neutrophils, we found that the CD31 agonist consistently sustained CD31 phosphorylation (*Figure 4—figure supplement 1A*) and clustering (*Figure 4—figure supplement 1B*) during fMLP activation (*Figure 4—figure supplement 1C*), specifically targeting the CD31 molecule (*Figure 4—figure supplement 1D*), and fine-tuning cytoskeleton dynamics (*Figure 4—figure supplement 1E*). To investigate the involvement of CD31 in regulating uropod detachment in migrating neutrophils, we conducted single-cell-level analysis using a model of skin inflammation. To track neutrophils in vivo, we used two-photon microscopy and *Lyz2*-GFP⁺ mice. These mice received IL-1β in the ear skin, with or without the administration of the CD31 agonist to enhance CD31 signaling. Subsequently, we recorded the extravasation of *Lyz2*-GFP⁺ cells live, 1 or 3 hr after IL-1β injection. Our observations revealed that 1 hr after intradermal IL-1β injection, *Lyz2*-GFP⁺ cells exhibited crawling behavior along the post-capillary blood vessels. Extravasating leukocytes demonstrated delayed uropod detachment and underwent significant elongation before being disconnected from the basement membrane (*Figure 4A and B* and *Video 2*). These findings confirmed previously published results (*Hyun et al.,*

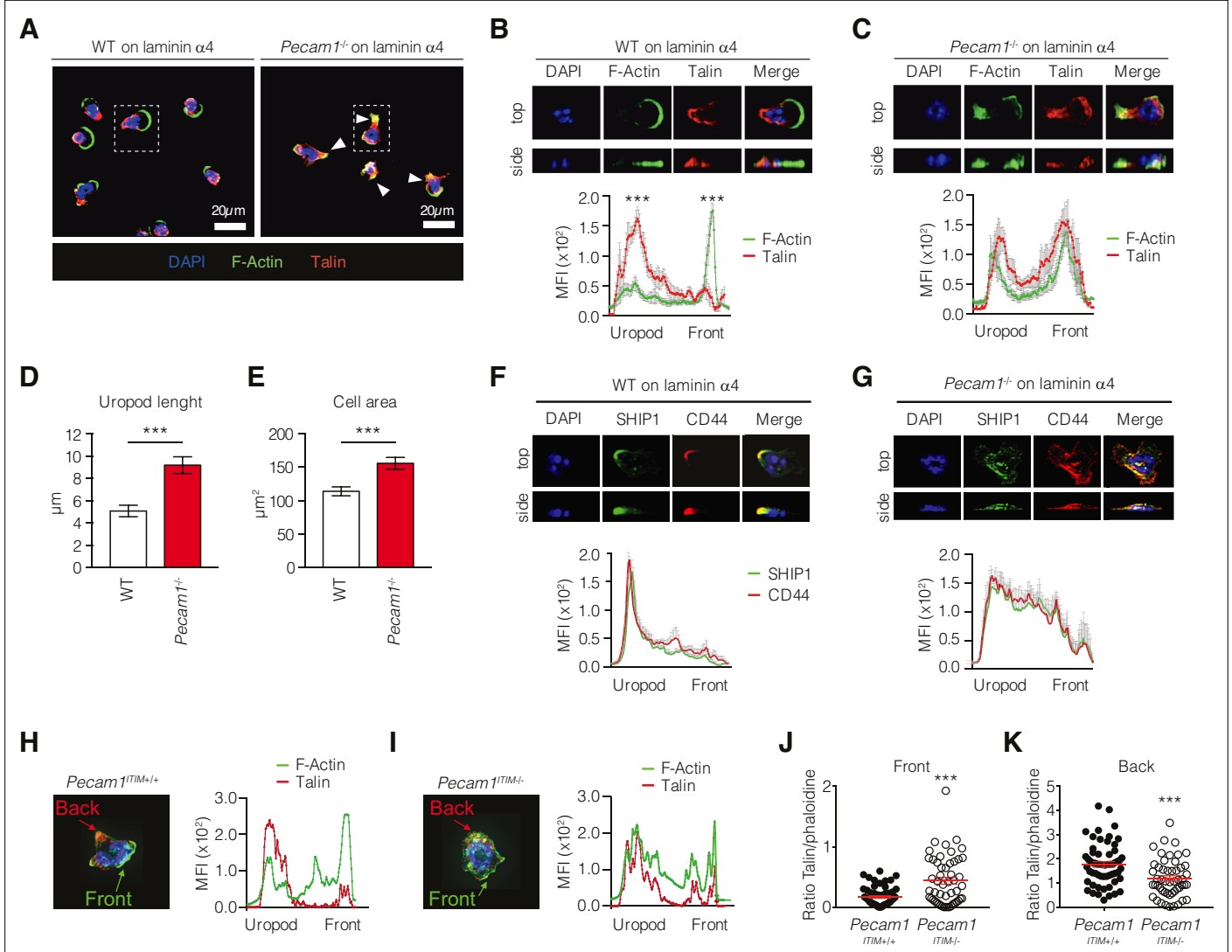

**Figure 3.** Phenotype of *Pecam1⁻/⁻* and ITIM-less CD31 neutrophils on laminin in vitro. (**A–G**) Confocal microscopy phenotypic characterization of purified WT or *Pecam1⁻/⁻* neutrophils placed on immobilized recombinant mouse laminin α4 and recombinant mouse CXCL1. (**A**) Representative confocal images and (**B, C**) quantification of talin (polyclonal, ab71333; red) and F-actin (Phalloidin, green) staining in WT mice (**B**) and *Pecam1⁻/⁻* mice (**C**). (**D**) Quantification of the uropod lengths of WT and *Pecam1⁻/⁻* neutrophils calculated as the distance from the backside of the nucleus to the end of the cellular tail. (**E**) Neutrophil total area. Data comes from three independent experiments, *p<0.05, **p<0.01, ***p<0.001 (unpaired Student's *t*-test). (**F, G**) Quantification of CD44 (clone IM7; red) and SHIP1 (clone PC1C; green) staining in WT mice (**F**) and *Pecam1⁻/⁻* mice (**G**). (**H, I**) Quantification of talin (red) and F-actin (Phalloidin, green) staining in *Pecam1*^ITIM+/+ mice (**H**) and *Pecam1*^ITIM-/- (**I**). Quantification of talin/phalloidin ratio at the front (**J**) and back (**K**) edge of analyzed cells. Quantification of mean fluorescence intensities (MFIs) of the indicated staining along the cell axis (mean ±SE [SE bars in gray color]; n > 8 cells/staining; statistical comparisons between the 20 first frontal and 20 last posterior measurement points are shown). *p<0.05, **p<0.01, ***p<0.001 (Mann–Whitney test).

The online version of this article includes the following source data and figure supplement(s) for figure 3:

**Source data 1.** Quantification of mean fluorescence intensities (MFIs) of polymerized F-actin (phalloidin) and Talin (polyclonal ab71333) along the cell axis of WT neutrophils (*Figure 3B*).

**Source data 2.** Quantification of mean fluorescence intensities (MFIs) of polymerized F-actin (phalloidin) and Talin (polyclonal ab71333) along the cell axis of *Pecam1⁻/⁻* neutrophils (*Figure 3C*).

**Source data 3.** Uropod length (μm) of WT and *Pecam1⁻/⁻* neutrophils migrating on laminin α4-coated surfaces (*Figure 3D*).

**Source data 4.** Cell area (μm²) of WT and *Pecam1⁻/⁻* neutrophils migrating on laminin α4-coated surfaces (*Figure 3E*).

**Source data 5.** Quantification of mean fluorescence intensities (MFIs) of CD44 (clone IM7) and SHIP-1 (clone PC1C) along the cell axis of WT neutrophils (*Figure 3F*).

*Figure 3 continued*

**Source data 6.** Quantification of mean fluorescence intensities (MFIs) of CD44 (clone IM7) and SHIP-1 (clone PC1C) along the cell axis of *Pecam1*$^{-/-}$ neutrophils (*Figure 3G*).

**Source data 7.** Quantification of mean fluorescence intensities (MFIs) of polymerized F-actin (phalloidin) and Talin (polyclonal ab71333) along the cell axis of WT *Pecam1*$^{ITIM-/-}$ neutrophils (*Figure 3H*).

**Source data 8.** Quantification of mean fluorescence intensities (MFIs) of polymerized F-actin (phalloidin) and Talin (polyclonal ab71333) along the cell axis of *Pecam1*$^{ITIM-/-}$ neutrophils (*Figure 3I*).

**Source data 9.** Quantification of polymerized F-actin (phalloidin) and Talin (polyclonal ab71333) ratio at the leading edge of WT and *Pecam1*$^{ITIM-/-}$ neutrophils (*Figure 3J*).

**Source data 10.** Quantification of polymerized F-actin (phalloidin) and Talin (polyclonal ab71333) ratio at the uropod of WT and *Pecam1*$^{ITIM-/-}$ neutrophils (*Figure 3K*).

**Figure supplement 1.** Generation of CD31 exons 13–14 floxed, ITIM-less mice.

**Figure supplement 1—source data 1.** RT-QPCR analysis showing the expression of the different CD31 exons in the spleen of WT or *Pecam1*$^{ITIM-/-}$ mice (*Figure 3—figure supplement 1B*).

**Figure supplement 1—source data 2.** Western blot of CD31 intracellular and extracellular staining of WT, *Pecam1*$^{-/-}$, and *Pecam1*$^{ITIM-/-}$ mice (*Figure 3—figure supplement 1D*).

**Figure supplement 1—source data 3.** RT-QPCR analysis showing the expression of the different CD31 exons in the lungs of WT or *Pecam1*$^{ITIM-/-}$ mice (*Figure 3—figure supplement 1C*).

**Figure supplement 1—source data 4.** Hematological parameters and blood formula of WT and *Pecam1*$^{ITIM-/-}$ mice (*Figure 3—figure supplement 1E*).

*2012*) and prompted us to investigate the effect of the CD31 agonist on this process. Administration of the CD31 agonist 1 hr after the initiation of *Lyz2*-GFP$^+$ cell extravasation resulted in a remarkable acceleration of uropod detachment from the basement membrane. This led to an increased migratory capacity of *Lyz2*-GFP$^+$ cells in the extravascular space compared to cells from vehicle-treated mice (*Figure 4A and B* and *Video 3*). Assessment of cell length and circularity provides valuable insights into their ability to migrate through narrow spaces. Cells with longer lengths and a higher surface area to volume ratio have an advantage in navigating tight spaces and interacting with the extracellular matrix, enabling efficient movement within confined environments. By evaluating the length and circularity of the cells, we aimed to assess their proficiency in migration through narrow spaces and their ability to maneuver through the extracellular matrix. Furthermore, our investigation of the fate of extravasated *Lyz2*-GFP$^+$ cells 3 hr after inducing inflammation revealed that the CD31 agonist enhanced their migratory capacity in the extravascular space (*Figure 4C and D* and *Videos 4 and 5*). This enhancement was demonstrated by increased speed, shorter cellular length, and greater traveling distance compared to the control group (*Figure 4E and F*). Taken together, these gain-of-function experiments provided further evidence that CD31 signaling actively participates in uropod detachment from the vessel after neutrophil transendothelial migration outside the vessel wall.

## Integrins inhibition rescues the *Pecam1*$^{-/-}$ neutrophil migration defects in vivo

Since functional CD31 and inactive β2-integrin colocalize at the uropod under chemotactic conditions, we hypothesized that CD31 could be involved in coordinating uropod detachment by reducing the integrin adhesive state. We reasoned that, if the impaired migration in *Pecam1*$^{-/-}$ neutrophils resulted from defective control of integrin adhesion, reducing integrin interactions with the extracellular adhesive surfaces would have compensated for the absence of CD31. To experimentally test this hypothesis, we designed an in vivo rescue experiment using soluble RGD peptides to antagonize integrin binding to the ECM (*Figure 5A*). When RGD peptides are administered as soluble, they act as a decoy for the integrins binding site, preventing leukocyte retention on ECM surfaces, thereby promoting cellular forward movements (*Mondal et al., 2012*). In WT mice, RGD peptides did not modify IL-1β-induced neutrophil accumulation in the peritoneal cavity (*Figure 5B and C*). However, extravascular (i.p.) injection of RGD peptides rescued the defective neutrophil accumulation in *Pecam1*$^{-/-}$ mice upon IL-1β-induced peritonitis (*Figure 5B and C*). Corresponding fluorescent microscopy analysis confirmed the enhanced extravascular detachment of *Pecam1*$^{-/-}$ neutrophils upon RGD injection (*Figure 5D and E*). Based on the observation that the CD31 agonist increased neutrophil recruitment into the peritoneal cavity (*Figure 5F–H*), we investigated whether CD31 intracellular signaling, rather than its

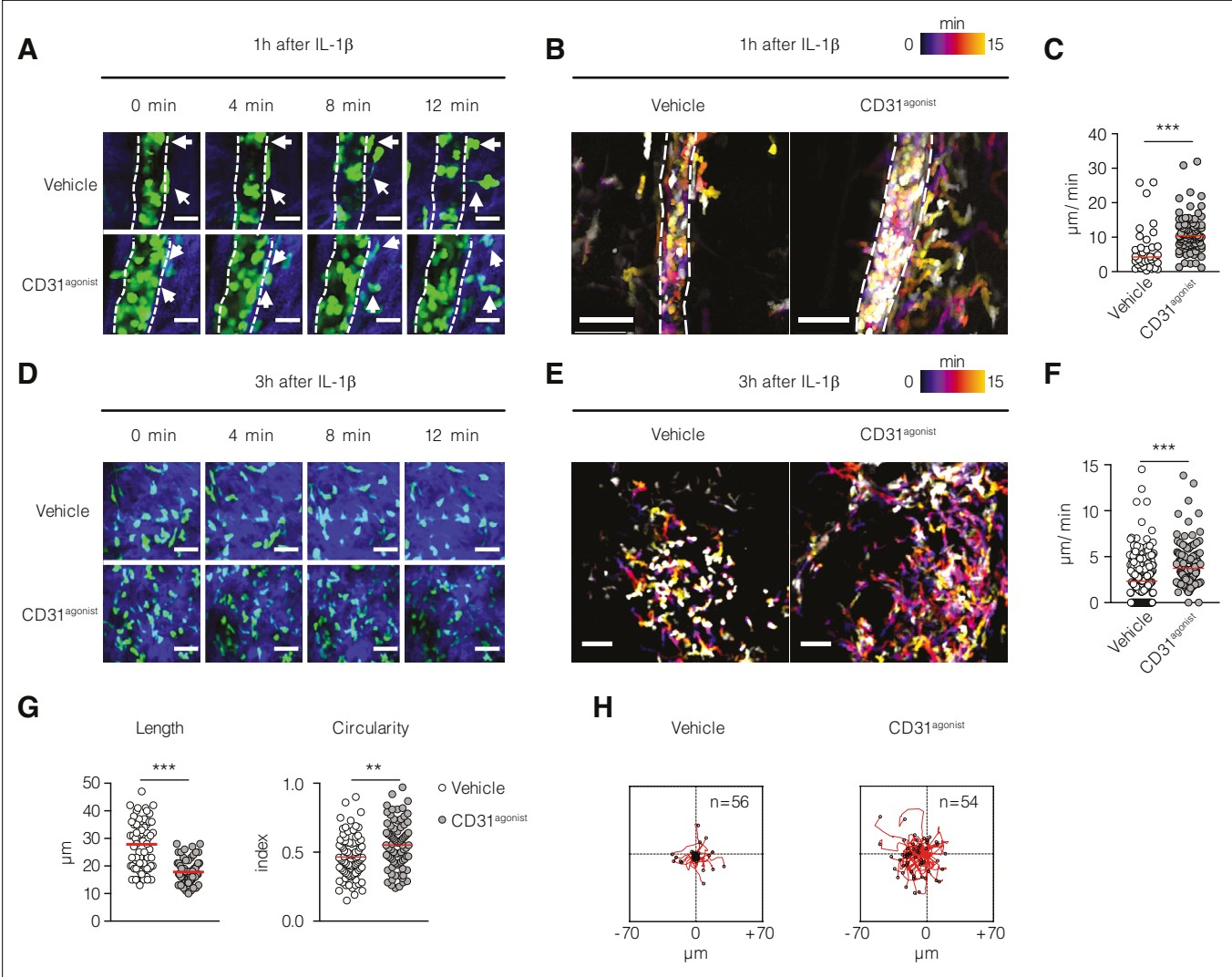

**Figure 4.** Effect of CD31 engagement on neutrophil detachment from the vessel wall and recruitment in vivo. In vivo multiphoton intravital microscopy of *Lyz2*-GFP+ mice ear pinna intradermally injected with IL-1β with or without the CD31 agonist (*Video 5*). (**A**) Post-capillary venules at 1 hr after IL-1β injection in *Lyz2*-GFP+ mice show GFP+ neutrophils (in green) that are extravasating (white arrows). Blue color represents the collagen fibers imaged with the secondary harmonic generation (SHG). Scale bar 20 μm. (**B**) Cell locomotion during time is visualized by time color-coded superposition of frames. Scale bar 50 μm. (**C**) Quantification of detachment speed of individual neutrophils from the vessel wall. Data are expressed as mean showing all points. ***p<0.001 (Mann–Whitney test). n = 34–90 cells pulled from two mice per condition. (**D**) Extravasated leukocytes at 3 hr after IL-1β injection. Scale bar 40 μm. (**E**) Time color-coded superposition of frames at 3 hr after IL-1β injection. Scale bar 40 μm. (**F**) Quantification of migratory speed at 3 hr after IL-1β injection. (**G**) Morphological parameters (length and circularity) of leukocytes at 3 hr after IL-1β injection. Data are expressed as mean showing all points. **p<0.01, ***p<0.001 (Mann–Whitney test). n = 60 cells pulled from two mice per condition. (**H**) Track plots of *Lyz2*-GFP+ cells at 3 hr after IL-1β injection treated with vehicle (left) and treated with the CD31^agonist (right) normalized to the starting position.

The online version of this article includes the following source data and figure supplement(s) for figure 4:

**Source data 1.** Detachment speed of neutrophils egressing from the post-capillary venules 1 hr after IL-1β challenge in the ear pinna of Lyz2-GFP mice treated or not with CD31 agonist (*Figure 4C*).

**Source data 2.** Migratory speed of neutrophils 3 hr after IL-1β challenge in the ear pinna of Lyz2-GFP mice treated or not with CD31 agonist (*Figure 4F*).

**Source data 3.** Morphological parameters of neutrophils 3 hr after IL-1β challenge in the ear pinna of Lyz2-GFP mice treated or not with CD31 agonist (*Figure 4G*).

**Source data 4.** Track plots of Lyz2-GFP+ cells at 3 hr after IL-1β injection treated with vehicle (*Figure 4H*, left).

**Source data 5.** Track plots of Lyz2-GFP+ cells at 3 hr after IL-1β injection treated with CD31 agonist (*Figure 4H*, right).

**Figure supplement 1.** CD31 agonist peptide is able to sustain CD31 functionality in human and murine neutrophils.

*Figure 4 continued on next page*

*Figure 4 continued*

**Figure supplement 1—source data 1.** Dynamic binding of the CD31 agonist (FITC-conjugated) on human neutrophil under basal and activated conditions analyzed at the indicated time points by flow cytometry (*Figure 4—figure supplement 1C*).

**Figure supplement 1—source data 2.** Western blot of phospho-ITIM CD31 in human neutrophils treated with fMLP with incremental doses of CD31 agonist (*Figure 4—figure supplement 1A*).

**Figure supplement 1—source data 3.** Quantification of CD31 agonist binding on WT and *Pecam1*[-/-] neutrophils (*Figure 4—figure supplement 1D*).

**Figure supplement 1—source data 4.** Quantification of actin polymerization (Phalloidin staining) upon fMLP challenging the presence or absence of CD31 agonist (*Figure 4—figure supplement 1E*).

ectodomain properties, plays a role in regulating the recruitment of neutrophils to the site of inflammation. To explore this, we conducted an adoptive transfer experiment, injecting a 1:1 suspension of WT and *Pecam1*[ITIM-/-] neutrophils into WT recipient mice. Prior to injection, cells were stained with CTV or CFSE, respectively, to track their origin. After 24 hr, the recipient mice were intraperitoneally challenged with IL-1β, and the transmigrated neutrophils were collected from the peritoneal wash 4 hr later in the competition assay (*Figure 5I*). Despite equal frequencies of WT and *Pecam1*[ITIM-/-] neutrophils (identified by flow cytometry as CD45[+] Ly6G[+]) in the blood before and after IL-1β injection (*Figure 5J and K*), *Pecam1*[ITIM-/-] neutrophils exhibited a disadvantage in recruitment into the inflamed peritoneum compared to their WT counterparts.

In summary, our collective findings provide compelling evidence that the CD31 ITIM-dependent intracellular pathway plays a pivotal role in regulating neutrophil migration within the extravascular space during inflammation. Specifically, it effectively controls integrin activation at the uropod, thereby exerting a significant influence on the efficient movement of neutrophils. This regulatory mechanism ensures the timely arrival of neutrophils at the inflammatory site, facilitating their efficient migration and precise targeting of the inflamed area, thereby optimizing the immune response.

## Discussion

Our study uncovered a significant and previously unknown role for CD31 in multiple stages of neutrophil migration and recruitment to sites of inflammation. We found that CD31 signaling at the uropod plays a key role in the polarization and detachment of neutrophils, which is crucial for initiating their migration within the vascular compartment and subsequent progression through the interstitial

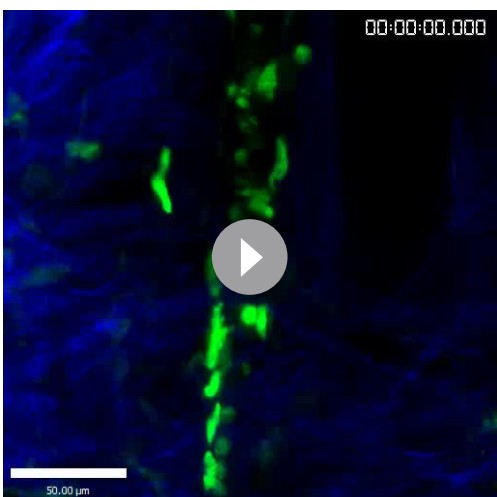

**Video 2.** Behavior of *Lyz2*-GFP[+] neutrophils in the ear pinna after 1 hr of IL-1β challenge. Example of post-capillary venule at 1 hr after IL-1β injection in *Lyz2*-GFP[+] mice showing neutrophils (in green) that are extravasating. Blue color represents the collagen fibers imaged with the secondary harmonic generation (SHG). Scale bar represents 50 µm.

https://elifesciences.org/articles/84752/figures#video2

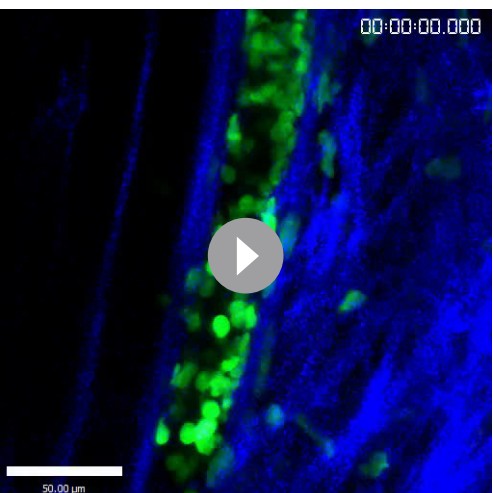

**Video 3.** Behavior of CD31[agonist]-treated *Lyz2*-GFP[+] neutrophils in the ear pinna after 1 hr of IL-1β challenge. Example of post-capillary venule at 1 hr after IL-1β injection in *Lyz2*-GFP[+] mice injected with CD31 agonist. Scale bar represents 50 µm.

https://elifesciences.org/articles/84752/figures#video3

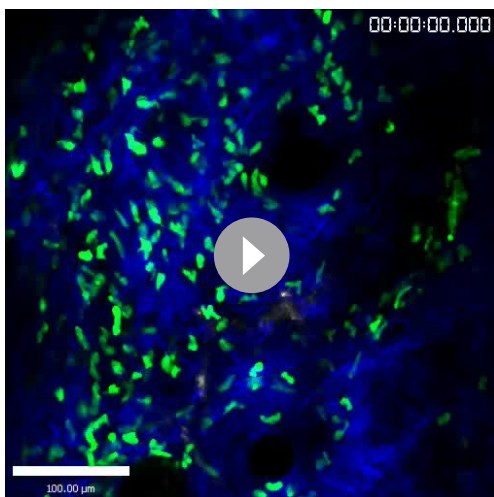

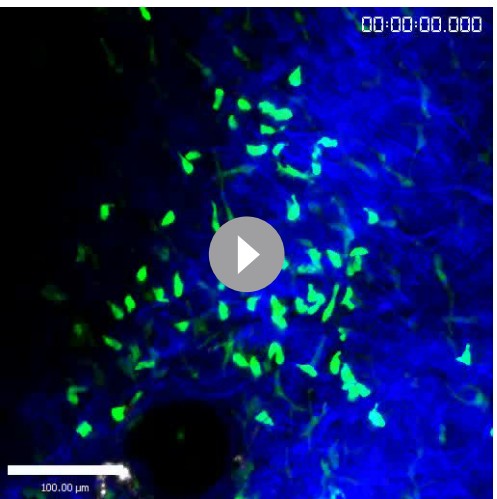

**Video 4.** Behavior of CD31^agonist-treated *Lyz2*-GFP⁺ neutrophils in the ear pinna after 3 hr of IL-1β challenge. Example of migrating *Lyz2*-GFP⁺ neutrophils (in green) 3 hr after IL-1β injection. Scale bar represents 100 µm.

https://elifesciences.org/articles/84752/figures#video4

**Video 5.** Behavior of *Lyz2*-GFP⁺ neutrophils in the ear pinna after 3 hr of IL-1β challenge. Example of migrating *Lyz2*-GFP⁺ neutrophils (in green) 3 hr after IL-1β injection. Scale bar represents 100 µm.

https://elifesciences.org/articles/84752/figures#video5

tissue following extravasation towards inflammatory sites. Our observations confirmed the dynamic behavior of neutrophils, wherein the uropod remains trapped at the basolateral membrane of endothelial cell junctions, while the leading edge moves forward in the interstitial tissue (*Hind et al., 2016*). The detachment of the uropod is a crucial process that enables neutrophils to transition from a firmly adhesive state within blood vessels, under strong hemodynamic conditions, to a rapidly migratory configuration in the static extravascular space. This transition is characterized by reduced adhesion and increased contractility, which facilitate effective navigation throughout interstitial tissue. Although our study highlights the importance of CD31 signaling in uropod detachment, we did not specifically investigate the molecular events involved in the interaction between CD31 and ICAM-1 or laminin α4, which are RGD-binding-independent interactions in the vascular compartment. Further investigations focusing on the closure of these integrins, such as αMβ2 with ICAM-1 and α6β1 with laminin α4, are required to understand the specific interactions between CD31-positive cells and the matrix in the vascular compartment. Nevertheless, our data unequivocally demonstrated that the administration of soluble RGD peptides in the extravascular space effectively rescued the stagnation of *Pecam1*⁻/⁻ neutrophils within the extracellular matrix. These peptides competitively bind to the RDG motif found in various extracellular matrix proteins, potentially reversing the excessive adhesion of RGD-binding β1 integrins and facilitating proper neutrophil migration toward sites of inflammation. Therefore, our data strongly suggest that the retention of *Pecam1*⁻/⁻ neutrophils in the extravascular interstitium is primarily due to the excessive adhesion of integrins to the components of the extracellular matrix. This heightened integrin adhesion hinders the initiation of neutrophil movement through the interstitial tissue toward inflammatory sites. Previous studies have proposed that impaired CD31-dependent upregulation of β1-integrin leads to the inability of *Pecam1*⁻/⁻ neutrophils to exit blood vessels (*Dangerfield et al., 2002*). However, recent evidence challenges this hypothesis by demonstrating that neutrophils lacking β1-integrins actually have an enhanced ability to reach inflamed sites (*Sarangi et al., 2012*). Our present findings suggest that CD31 may be involved in β1-integrin conformation rather than solely regulating its expression. Further investigations are needed to comprehensively elucidate the molecular mechanisms underlying CD31's regulatory functions on β1-integrin behavior during neutrophil migration.

While our primary focus was on investigating CD31's role in uropod detachment, proteomic analysis revealed that CD31 partners were also associated with the actin cytoskeleton. Moreover, we observed that the same phenotype was obtained in two scenarios, namely when CD31 expression was completely absent and when the protein was present but a part of its intracellular tail containing the

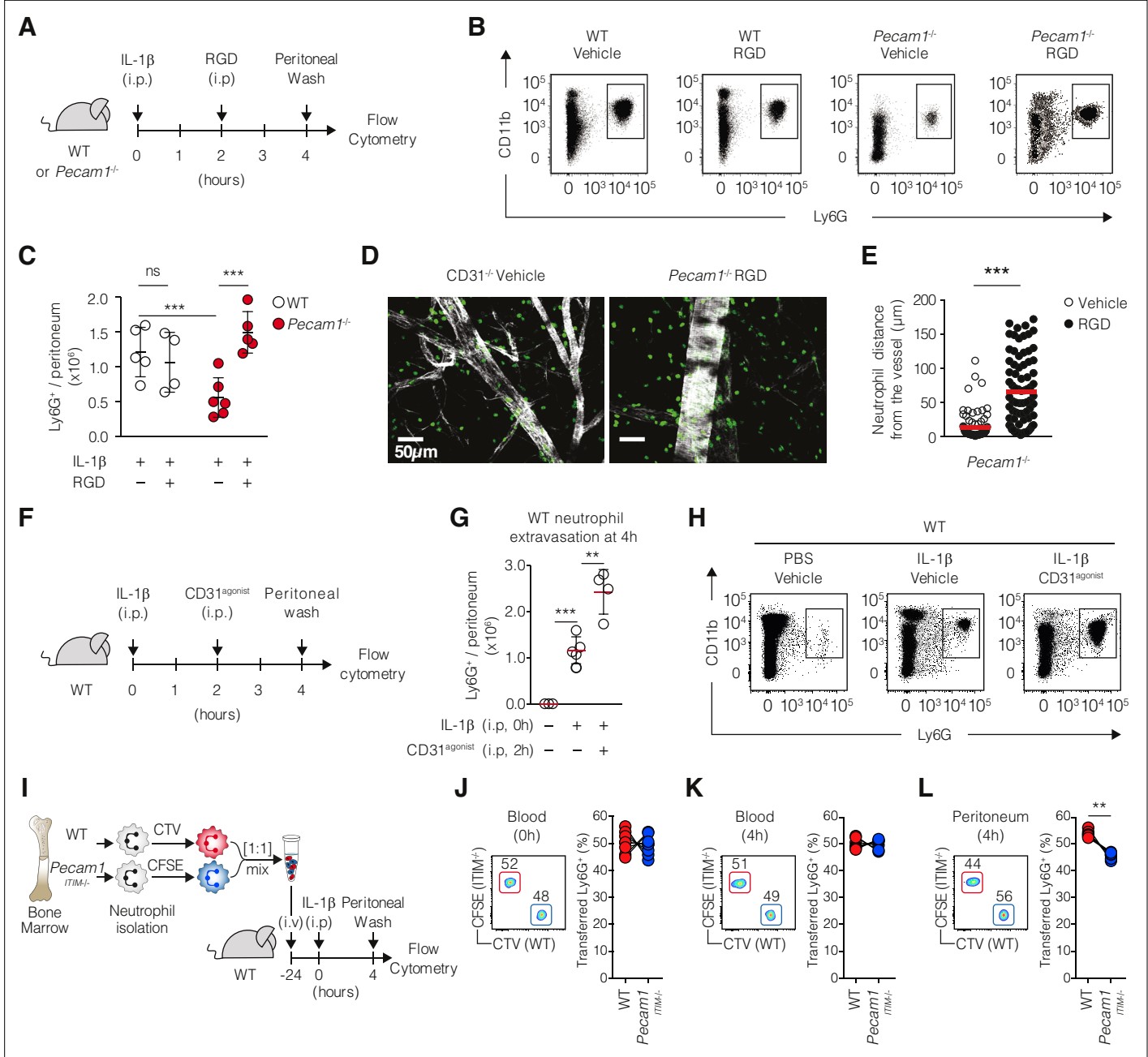

**Figure 5.** Restoring *Pecam1*[-/-] neutrophil motility in vivo through pharmacological interventions. (**A**) Experimental design. WT or *Pecam1*[-/-] mice (4–6/group) were injected intraperitoneally (i.p) with IL-1β to induce the peritonitis. After 2 hr, mice were injected i.p. with vehicle or RGD peptides. At 4 hr, mice were sacrificed, and absolute peritoneal neutrophil number was assessed. (**B**) Representative dot plots showing harvested cellular exudate at 4 hr in the indicated conditions. Among CD45[+] cells, neutrophils were identified as CD11b[+] and Ly6G[+] by flow cytometry. (**C**) Absolute numbers of extravasated neutrophils at 4 hr in the indicated conditions. (**D**) Confocal micrographs of whole-mount omentum showing extravasating Ly6G[+] cells (green) from a post-capillary venule (Phalloidin, white) in *Pecam1*[-/-] mice. Scale bar 50 μm. (**E**) Quantification of *Pecam1*[-/-] neutrophil distance from the closest post-capillary vessels in vehicle or RGD-treated mice. (**F**) Experimental design. WT (4–5/group) were injected i.p with or without IL-1β to induce the peritonitis. After 2 hr, mice were injected i.p. with vehicle or CD31[agonist]. At 4 hr, mice were sacrificed, and absolute peritoneal neutrophil number was assessed. (**G**) Representative dot plots and (**H**) absolute numbers of extravasated neutrophils at 4 hr in the indicated conditions. Data are expressed as mean ± SD. Mann–Whitney test *p<0.05, **p<0.01, ***p<0.001. (**I**) Experimental setup. WT or *Pecam1*[ITIM-/-] neutrophils were purified from the bone marrow, differentially stained with CTV or CFSE, and mixed in 1:1 ratio. Cells were adoptively transferred into WT recipient. After 24 hr, recipient mice were challenged with intraperitoneal injection of IL-1β, and transmigrated neutrophils were analyzed in the peritoneal wash after 4 hr. Neutrophils were identified as live, singlets, CD45[+], CD11b[+], and Ly6G[+] cells. (**J, K**) Representative FACS plots (left) and proportion (right) of WT and *Pecam1*[ITIM-/-]

*Figure 5 continued on next page*

*Figure 5 continued*

neutrophils in the blood before (**J**) and 4 hr after (**K**) IL-1β challenge. (**L**) Representative FACS plots (left) and proportion (right) of WT and *Pecam1*[ITIM-/-] neutrophils in the peritoneal wash 4 hr after IL-1β challenge. N = 7, **p<0.01 (Wilcoxon matched-pairs signed-rank test).

The online version of this article includes the following source data for figure 5:

**Source data 1.** Absolute numbers of extravasated neutrophils at 4 hr in WT and *Pecam1*[-/-] mice treated or not with RGD peptides (***Figure 5C***).

**Source data 2.** Quantification of *Pecam1*[-/-] neutrophil distance from the closest post-capillary vessels in vehicle or RGD-treated mice (***Figure 5E***).

**Source data 3.** Absolute numbers of extravasated neutrophils at 4 hr in the indicated conditions (***Figure 5G***).

**Source data 4.** Proportion of transferred WT or *Pecam1*[ITIM-/-] neutrophils in the blood before IL-1β challenge (***Figure 5J***).

**Source data 5.** Proportion of transferred WT or *Pecam1*[ITIM-/-] neutrophils in the blood 4 hr after IL-1β challenge (***Figure 5K***).

**Source data 6.** Proportion of transferred WT or *Pecam1*[ITIM-/-] neutrophils in the peritoneal fluid 4 hr after IL-1β challenge (***Figure 5L***).

ITIM motifs was deleted. This indicates that both the presence of CD31 and its intact intracellular tail with the ITIM motifs are essential for proper functional cell polarization during neutrophil migration. This polarization defect could potentially impact the coordination between the leading edge and the uropod, which is crucial for effective neutrophil extravasation (***Hind et al., 2016***).

After being phosphorylated by active co-clustered ITAM receptors, the ITIM motifs of CD31 act as docking sites for several proteins (***Henshall et al., 2001***; ***Sagawa et al., 1997***) and lipid SH2-containing phosphatases (***Ilan and Madri, 2003***, p. 1), which drive downstream signaling pathways upon CD31 engagement in various biological processes. Previous studies have highlighted the significant involvement of lipid phosphatases in the regulation of leukocyte migration (***Michael et al., 2021***; ***Mondal et al., 2012***; ***Nishio et al., 2007***). Our data strongly suggest that SHIP-1 acts as a critical mediator of CD31 signaling in integrin closure during neutrophil migration. Further investigations specifically targeting the type of phosphatase involved in CD31-dependent uropod detachment and polarization are warranted to gain a more comprehensive understanding of the underlying mechanisms.

The concentration of endothelial CD31 at lateral borders (***Newman, 1994***) likely plays a crucial role in regulating the engagement of CD31 on neutrophils during transendothelial migration. Moreover, CD31 undergoes rapid tyrosine phosphorylation in response to mechanical stress (***Masuda et al., 2004***; ***Snyder et al., 2017***). These findings suggest that, as neutrophils pass through the lateral endothelial cell junctions, CD31 may sense the gradient of mechanical forces generated within the cell across the vascular barrier. The cell experiences stretching due to firm intravascular adhesion, counteracting shear stress, and easy movement of the extravascular leading edge in the unloaded static environment of the extravascular space. The magnitude of this gradient is reflected by the elongation of the uropod out of the basal membrane. Based on our observations, we propose that the concentration of CD31 at the uropod may be required to 'free' this pole of the cell, allowing it to continue its journey toward the inflammatory site guided by chemokine gradients in the extravascular space. This freeing action, promoted by CD31, is potentially mediated by the recruitment of phosphatases to its mechanically driven phosphorylated ITIMs, resulting in a closed/inactive conformation of integrins at the uropod (***Figure 2J***). These observations position CD31 as a potential molecular sensor involved in regulating integrin conformation and facilitating the transition from neutrophil egression to migration.

In conclusion, our study provides important insights into the multifaceted role of CD31 in neutrophil migration and extravasation. Further investigations into the specific interactions and downstream signaling events involving CD31 and its partners will deepen our understanding of neutrophil migration and may pave the way for novel therapeutic interventions for inflammatory diseases.

# Materials and methods

**Key resources table**

| Reagent type (species) or resource | Designation | Source or reference | Identifiers | Additional information |
|---|---|---|---|---|
| Antibody | Anti-mouse CD31, rat monoclonal (clone 390) | BD Biosciences | Cat# 558736, RRID:AB_397095 | 1:100 |

*Continued on next page*

*Continued*

| Reagent type (species) or resource | Designation | Source or reference | Identifiers | Additional information |
|---|---|---|---|---|
| Antibody | Anti-mouse CD31, rabbit monoclonal (clone SP38) | Abcam | Cat# ab231436 | 1:100 |
| Antibody | Anti-human CD31, mouse monoclonal (clone MBC78.2) | Merk | Cat# MABF2034 | 1:100 |
| Antibody | APC anti-human CD11b, mouse monoclonal (clone D12) | BD Biosciences | Cat# 561015, RRID:AB_10561676 | 1:100 |
| Antibody | Alexa Fluor 647 anti-human CD11a, mouse monoclonal (clone NKI-L16) | Thermo Fisher | Cat# MUB0364P | 1:100 |
| Antibody | PE anti-human CD11a, mouse monoclonal (clone TS2/4) | BioLegend | Cat# 350605, RRID:AB_10660819 | 1:100 |
| Antibody | Alexa Fluor 647 anti-human SHIP-1, mouse monoclonal (clone P1C1-A5) | BioLegend | Cat# 656608, RRID:AB_2563145 | 1:100 |
| Antibody | Alexa Fluor 488 anti-human CD31, mouse monoclonal (clone WM59) | BioLegend | Cat# 303109, RRID:AB_493075 | 1:100 |
| Antibody | FITC anti-mouse CD45, rat monoclonal (clone 30-F11) | BD Biosciences | Cat# 553080, RRID:AB_394610 | 1:100 |
| Antibody | PE anti-mouse Ly6G, rat monoclonal (clone 1A8) | BD Biosciences | Cat# 551461, RRID:AB_394208 | 1:100 |
| Antibody | APC anti-mouse CD11b, rat monoclonal (clone M1/70) | BD Biosciences | Cat# 553312, RRID:AB_398535 | 1:100 |
| Antibody | Alexa Fluor 647 AffiniPure anti-rat IgG, (H+L), donkey polyclonal | Jackson ImmunoResearch | Cat# 712-605-153, RRID:AB_2340694 | 1:500 |
| Antibody | Alexa Fluor 594 AffiniPure anti-goat IgG (H+L), donkey polyclonal | Jackson ImmunoResearch | Cat# 705-585-147, RRID:AB_2340433 | 1:500 |
| Antibody | Alexa Fluor 488 AffiniPure anti-rabbit IgG (H+L), donkey polyclonal | Jackson ImmunoResearch | Cat# 711-545-152, RRID:AB_2313584 | 1:500 |
| Antibody | Anti-CD31 (phospho Y713) antibody, rabbit monoclonal (clone EPR8079(2)) | Abcam | Cat# ab180175 | 1:100 |
| Antibody | Anti-mouse/rat CD31, goat polyclonal | R&D Systems | Cat# AF3628, RRID:AB_2161028 | 1:100 |
| Antibody | Anti-mouse/human Talin 1, rabbit polyclonal | Abcam | ab71333, RRID:AB_2204002 | 1:100 |
| Antibody | Anti-mouse SHIP-1, mouse monoclonal (clone P1C1-A5) | BioLegend | Cat# 656601, RRID:AB_2562400 | 1:100 |
| Antibody | Alexa Fluor(R) 488 anti-mouse/human CD44, rat monoclonal (clone IM7) | BioLegend | Cat# 103016, RRID:AB_493679 | 1:100 |
| Antibody | Anti-human GAPDH, rabbit polyclonal | Sigma-Aldrich | Cat# G9545, RRID:AB_796208 | 1:1000 |
| Antibody | Anti-rabbit IgG, HRP-conjugate antibody, giat polyclonal | Millipore | Cat# 12-348, RRID:AB_390191 | 1:2000 |
| Peptide, recombinant protein | Recombinant human IL-8/CXCL8 protein | R&D Systems | Cat# 208-IL-050 | 10 ng/ml |

*Continued on next page*

*Continued*

| Reagent type (species) or resource | Designation | Source or reference | Identifiers | Additional information |
|---|---|---|---|---|
| Peptide, recombinant protein | Recombinant human ICAM-1 | R&D Systems | Cat# ADP4 | 20 µg/ml |
| Peptide, recombinant protein | Recombinant mouse laminin a4 | R&D Systems | Cat# 3837-A4 | 20 µg/ml |
| Peptide, recombinant protein | Recombinant mouse CXCL1 | R&D Systems | Cat# 453-KC | 10 ng/ml |
| Peptide, recombinant protein | Recombinant mouse IL-1 beta | R&D Systems | Cat# 401ML-005 | 40 ng/ml |
| Peptide, recombinant protein | RGD peptide | Tocris | Cat# 3498 | 10 mg/kg |
| Peptide, recombinant protein | CD31 agonist (P8RI) | PMID:29957231 | N/A | 2.5 mg/kg |
| Peptide, recombinant protein | N-Formyl-Met-Leu-Phe | Tocris | Cat# 1921 | 1 µM |
| Chemical compound, drug | Viobility 405/520 fixable dye | Miltenyi Biotec | Cat# 130-109-816 | 1:100 |
| Chemical compound, drug | Alexa Fluor 647 Phalloidin | Thermo Fisher Scientific | Cat# A22287, RRID:AB_2620155 | 150 nM |
| Chemical compound, drug | DAPI | Thermo Fisher Scientific | Cat# D1306; RRID:AB_2629482 | 5 µM |
| Chemical compound, drug | Percoll | Sigma | Cat#P1644 | |
| Commercial assay or kit | ProLong Gold Antifade Mountant | Invitrogen | Cat# P36930 | |
| Commercial assay or kit | CellTrace Violet Cell Proliferation Kit | Thermo Fisher Scientific | Cat# C34571 | |
| Commercial assay or kit | CellTrace CFSE Cell Proliferation Kit | Thermo Fisher Scientific | Cat# C34554 | |
| Commercial assay or kit | EasySep Mouse Neutrophil Enrichment Kit | Stem Cell Technologies | Cat# 19762 | |
| Commercial assay or kit | Mouse CXCL1/KC DuoSet ELISA | R&D Systems | Cat# DY453-05 | |
| Commercial assay or kit | Bio-Plex Pro Magnetic COOH Beads 26 | Bio-Rad | Cat# MC10026-01 | |
| Commercial assay or kit | Bio-Plex Amine Coupling Kit | Bio-Rad | Cat# 171406001 | |
| Commercial assay or kit | Pierce BCA Protein Assay Kit | Thermo Scientific | Cat# 23225 | |
| Commercial assay or kit | 10% Mini-PROTEAN TGX Precast Protein Gels, 10-well, 50 µl | Bio-Rad | Cat# 4561034 | |
| Commercial assay or kit | Trans-Blot Turbo Mini 0.2 µm Nitrocellulose Transfer Packs | Bio-Rad | Cat# 1704158 | |
| Commercial assay or kit | Clarity Western ECL substrate kit | Bio-Rad | Cat# 1705060S | |
| Strain, strain background (*Mus musculus*) | C57BL/6, C57BL/6 | Charles River | Internal colony | |
| Strain, strain background (*M. musculus*) | *Pecam1*$^{-/-}$, C57BL/6 | PMID:10072554 | Internal colony | |
| Strain, strain background (*M. musculus*) | *Lyz2*-eGFP, C57BL/6 | PMID:10887140 | Internal colony | |
| Strain, strain background (*M. musculus*) | *Pecam1*$^{ITIM-/-}$, C57BL/6 | Generated for this work | Internal colony | |

*Continued on next page*

*Continued*

| Reagent type (species) or resource | Designation | Source or reference | Identifiers | Additional information |
|---|---|---|---|---|
| Software, algorithm | FlowJo V10 | FlowJo | https://www.flowjo.com/ | |
| Software, algorithm | Mascot | Matrix Sciences | http://www.matrixscience.com | |
| Software, algorithm | Enrichr | PMID:27141961 | https://maayanlab.cloud/Enrichr/ | |
| Software, algorithm | Prism 9 | GraphPad Software | https://www.graphpad.com/scientific-software/prism | |
| Software, algorithm | BD FACSDiva V8 | BD Biosciences | https://www.bdbiosciences.com/en-us/products/software/instrument-software/bd-facsdiva-software | |
| Software, algorithm | Leica Application Suite X (LAS X) | Leica Microsystems | https://www.leica-microsystems.com/products/microscope-software/p/leica-las-x-ls/ | |
| Software, algorithm | Ingenuity Pathway Analysis (IPA) | QIAGEN | https://www.qiagen.com | |
| Software, algorithm | Fiji-ImageJ | ImageJ | https://imagej.net/Fiji/ | |
| Other | FACS LSR II | BD Biosciences | N/A | Flow cytometer |
| Other | LTQ Velos Orbitrap | Thermo Fisher Scientific | N/A | Mass spectrometer |
| Other | Zeiss LSM7 MP | Zeiss | N/A | Fluorescent microscope |
| Other | AxioObserver | Zeiss | N/A | Fluorescent microscope |
| Other | Trans-Blot Turbo Transfer System | Bio Rad | N/A | Western blot transfer system |
| Other | Zeiss TIRF 3 system | Zeiss | N/A | TIRF microscope |
| Other | Vet ABC | scil | N/A | Hemocytometer |

## Isolation of primary human neutrophils

Peripheral blood was collected by venipuncture from healthy adult volunteers using potassium EDTA-coated vacutainers (BD Biosciences #367861). Blood (6 ml) was fractionated on discontinuous Percoll (Sigma #P1644) gradients consisting of two isotonic layers (w/o divalent cations, 3.5 ml each) of 1.081 g/ml and 1.092 g/ml. After centrifugation for 20 min at 540 × *g*, the interface between the plasma and the 1.081 g/ml layer contained the PBMC and platelets, the interface between the 1.081 g/ml and the 1.092 g/ml layer contained the granulocyte fraction, while erythrocytes are pelleted at the bottom of the tube. The granulocyte fraction was collected and washed twice in HBSS medium (w/o divalent cations, Gibco #14175095). All procedures were conducted at room temperature under sterile conditions. The preparations usually contained >97% granulocytes as determined by flow cytometry, with a cell viability >98% based on trypan blue exclusion.

## Mice

C56BL/6 WT mice were purchased from Charles River France Laboratories (L'Arbresle, France). Breeding pairs of *Pecam1*[-/-] mice on the C56BL/6 background were kindly provided by Dr. D. K. Newman (Milwaukee Blood Center, Milwaukee, WI). Mice were maintained in the animal facility under an alternating 12 hr light and 12 hr dark cycle in microisolator cages (≤4 per cage) under specific pathogen-free conditions with ad libitum access to food and water.

All the investigations conformed to the directive 2010/63/EU of the European Parliament, and formal approval was granted by the local Animal Ethics Committee (Comité d'étique Bichat-Debré, Paris, France).

*Lyz2*-eGFP mice were a kind gift from Professor Sussan Nourshargh (Queen Mary University of London). These mice have the gene for eGFP knocked into the Lysozyme (Lyz) M locus, resulting in mice with fluorescent myelomonocytic cells, with neutrophils comprising the highest percentage of eGFP[hi] cells. The mice were bred in-house (Central Research Facility, University of Glasgow) and

randomly assigned to experimental groups. All the procedures were performed in accordance with local ethical and UK Home Office regulations.

## Generation of CD31 exons 13–14 floxed mice (*Pecam1*$^{ITIM-/-}$)

bMQ382c16 BAC was modified in order to insert a neomycin (Neo) cassette flanked by FRT sites between exon 12 and exon 13 of the *Pecam1* gene, a first LoxP site between the exon 12 and the FRT-flanked Neo cassette, and a second LoxP site between exon 14 and exon 15 (*Figure 3—figure supplement 1*). The modified BAC was linearized and electroporated into CK35 ES cells. EC cell clones carrying the mutated *Pecam1* gene were selected by Southern blot and injected into C57BL/6J blastocytes. Mutant mice (carrying the neo(13/14) allele) were first crossed to FLP-recombinase-expressing mice in order to remove the Neo cassette and obtain mice carrying the lox(13-14) allele. Lox(13-14) mice were then crossed to transgenic mice expressing the Cre recombinase under the control of a CMV promoter in order to remove exons 13 and 14 (allele out(13-14), hereafter referred to as ITIM$^{-/-}$). This does not introduce a shift in the reading frame. Furthermore, as the depletion is ubiquitous, all further progenies can express the *Pecam1*-ITIM- allele even if the Cre is no longer expressed. Therefore, the *Pecam1*$^{ITIM+/-}$ CMV-Cre$^{+/-}$ animals were intercrossed, and *Pecam1*$^{ITIM+/-}$ negative for the CMV-Cre transgene were selected. *Pecam1*$^{ITIM+/-}$ mice were then intercrossed. Pups were born at the expected mendelian ratio, *Pecam1*$^{ITIM-/-}$ animals did not present any developmental abnormalities in comparison to their *Pecam1*$^{ITIM+/+}$ control littermates.

## Immunofluorescence staining of mouse tissues

Mice were sacrificed by intracardiac exsanguination under overdose of ketamine-HCl (150 mg/kg) and xylazine (30 mg/kg). Blood was collected in EDTA tubes for blood formula analysis. Mice were then perfused with PBS followed by paraformaldehyde (PFA) 3.7%. Lymph nodes and kidneys were collected and fixed in PFA 3.7%, then embedded in paraffin. Then, 4-μm-thick sections were deparaffinized in toluene and rehydrated in ethanol. Sections were incubated with retrieval reagent (R&D Systems) then immunostained using antibodies against CD31 extracellular Ig-like domain 2 (rat monoclonal; clone 390) or intracellular C-terminal (rabbit monoclonal, clone SP38) or isotype controls, followed by incubation with AF647-coupled F(ab')2 anti-rat IgG or AF647-coupled F(ab')2 anti-rabbit IgG. Nuclei were then stained with Hoechst 53542, and slides were mounted with fluorescent mounting medium (ProLong Gold, Thermo Fisher). Images were digitally captured using an AxioObserver epi-fluorescent microscope equipped with a Colibri 7 LED generator (Zeiss) and an ApoTome system and running Zen Software (Zeiss).

## Mass spectrometry

Peripheral blood neutrophils purified from one healthy donor were treated with or without fMLP (10$^{-6}$ M) for 20 min at 37°C, in duplicates. Cells were next lysed in a modified SDS-free RIPA buffer containing 1% Octyl-beta-d-glucopyranoside (Enzo Life Sciences #ALX-500-001) for 30 min on ice. Nuclei and cellular debris were separated from solubilized proteins by centrifugation (14,000 × *g*, 15 min at 4°C). Pre-cleared lysates were incubated with magnetic beads covalently bound with a monoclonal antibody directed against CD31 (clone MBC78.2) for 2 hr at 4°C. After washing with PBS-Tween (0.05%), bead-captured molecular complexes were digested with trypsin and processed for protein analysis with an LTQ Velos Orbitrap equipped with an EASY-Spray nanoelectrospray ion source coupled to an Easy nano-LC Proxeon 1000 system (Thermo Fisher Scientific). MS data were treated with Mascot search server for protein identification. Probability thresholds were set >90% probability for protein identifications, based upon at least two peptides identified with 80% certainty and with a protein coverage >20%. The score for each protein was normalized with the background score of the same protein obtained with a decoy immunoprecipitation in the same experimental conditions using beads coupled with a control isotype antibody. Normalized scores were accepted as significant if >10.

Proteins present in the CD31-pulldown were considered differentially expressed if their presence increased/decreased by 20% in each condition (cutoff: 0.8 and 1.2). N = 1 for each condition. The analysis of differentially associated proteins was performed with the core analysis module from the IPA system (http://www.ingenuity.com).

## Confocal and fluorescent microscopy of purified neutrophils

Polystyrene slides (ibidi #80826) were coated with recombinant human ICAM-1 (R&D Systems # ADP4) or recombinant mouse laminin α4 (R&D Systems #3837-A4) overnight at 20 µg/ml depending on the experiment. The next day, slides were blocked 1 hr with 1% BSA and recombinant human IL-8 (R&D Systems # 208-IL-050, 10 ng/ml) or recombinant mouse CXCL1 (R&D systems #453-KC, 10 ng/ml) for experiments involving human or mouse neutrophils, respectively. Freshly isolated human neutrophils or bone-marrow purified murine neutrophils (EasySep, Stemcell #19762) were placed on the respective coated surfaces for 5 min at 37°C. Cells were then fixed with 2% PFA for 10 min on ice and stained overnight at 4°C with purified primary antibodies (specified in the corresponding figure legend) in a buffer containing 0.2% saponin, 2.5% BSA, and 0.01% fish gelatin. The next day, samples were incubated with fluorescent-conjugated f(ab')$_2$ antibodies, counterstained with DAPI, and mounted in ProLong Gold antifade medium. Z-stack images (10 µm depth) were taken with a Zeiss Axiovert 200M inverted microscope equipped with an ApoTome module and post-treated so as to obtain a 3D reconstitution for each cell (n = 8/10 per condition). Quantification of mean fluorescence intensities (MFIs) of the indicated staining along the cell axis was performed with the ImageJ software using 8-bit images of the cellular top view (gray values range 0–255).

To assess the real-time distribution of CD31 in motile human neutrophils, purified cells were stained in suspension with fluorescent-conjugated antibodies directed against the CD31 (MBC78.2-PE) and CD11b (clone D12-APC) for 10 min at room temperature. Neutrophils were subsequently placed onto a fibronectin-coated surface (ibidi #80823) to allow their adherence/migration and videos were recorded for 10 min, starting from the addition of the neutrophil suspension to the wells using a Zeiss Axiovert 200M inverted microscope. Images were analyzed offline using the ImageJ software.

## Sterile peritonitis

Peritonitis was induced in WT and *Pecam1*$^{-/-}$ mice (9–13 weeks old, n = 3/6 per group) by the intraperitoneal (i.p.) administration of 1 ml containing 40 ng of IL-1β (R&D Systems #401) as described previously (*Thompson et al., 2001*). At each hour, mice were sacrificed, blood neutrophil percentage was evaluated with an automated cell counter (scil Vet ABC), and the peritoneal cavity was washed with 5 ml of PBS-EDTA (5 mM) in order to recover the cellular exudate. Cells were stained with monoclonal antibodies directed to CD45 (clone 30-F11), Ly6G (clone 1A8), CD11b (clone M1/70), and samples were analyzed by flow cytometry using an LSR II flow cytometer (BD Biosciences). Neutrophils were identified as CD45$^+$/CD11b$^+$/Ly6G$^+$ cells, and absolute cell count was determined using 123count eBeads (eBioscience #01-1234) according to the manufacturer's instructions. Plasma and cell-free peritoneal liquid of *Pecam1*$^{-/-}$ and wild-type (WT) mice were collected at 4 hr after IL-1β injection and CXCL1 concentration was evaluated by ELISA (R&D Systems #DY453). SDF-1, CD62P, and PTX-3 were evaluated by assessing a customized Luminex magnetic beads assay in which magnetic COOH beads (Bio-Rad #MC10026-01) were covalently immobilized with a monoclonal antibody directed against target molecules using a commercial coupling kit (Bio-Rad #171406001). In some experiments, mice received the intraperitoneal injection of RGD peptides (Arg-Gly-Asp-Ser, 10 mg/kg, 570 µM, 1 ml/peritoneal cavity. Tocris #3498) or the CD31$^{agonist}$ peptide (2.5 mg/kg, 50 µM, 1 ml/peritoneal cavity) at 2 hr after the induction of the peritonitis and cells were harvested at 4 hr.

## Omentum confocal microscopy

Confocal microscopy was performed on WT and *Pecam1*$^{-/-}$ dissected omental tissue fixed in 4% PFA for 1 hr at room temperature as previously described (*Buscher et al., 2016*). Briefly, tissues were blocked and permeabilized 1 hr at room temperature with a solution containing 0.5% TritonX, 5% BSA, and 0.1% fish gelatin. After extensive washes, samples were stained overnight at 4°C with purified monoclonal antibodies (specified in the figure legend) in PBS supplemented with 0.1% saponin. Detection was achieved using secondary f(ab')$_2$ antibodies coupled to fluorescent dyes (A488 and A647, Jackson ImmunoResearch). Parallel tissue samples were incubated without the primary antibodies in order to assess the specificity of the immunostainings (not shown). Omentum were finally counterstained with DAPI and mounted with ProLong Gold antifade medium (Invitrogen #P36930). Z-stacks (50 µm depth) were taken with a Zeiss Axiovert 200M inverted microscope equipped with an ApoTome module. Images were post-processed with the ImageJ software and presented as maximum-intensity projections. The intensity of CD31-pY713 was calculated in Ly6G-positive regions of interest (8-bit images,

grayscale values 0–255) and plotted against the perpendicular distance of the corresponding neutrophil from the closest post-capillary venule.

## In vivo multiphoton laser scanning microscopy (MPLSM)

*Lyz2*-GFP[+] mice (males, 10 weeks old) were anesthetized by ketamine/xylene injection and placed on the imaging stage at 37°C. Mice were injected in the ear pinna with 10 ng of IL-1β (R&D Systems #401ML-005) with or without the CD31[agonist] (50 µM). Mice were transferred immediately to the microscope stage and imaged for 15–20 min for each time point at an X-Y pixel resolution of 512 × 512. Multiphoton imaging was performed with a Zeiss LSM7 MP system equipped with both a ×10/0.3 NA air and ×20/1.0 NA water-immersion objective lens (Zeiss) and a tunable titanium/sapphire solid-state two-photon excitation source (Chameleon Ultra II; Coherent Laser Group; *Stephen et al., 2017*). Videos were processed offline with the ImageJ software for the generation of time-superposition images and the GFP[+] morphological parameters; the TrackMate plugin was implemented to calculate the speed of detaching neutrophils by applying a region of interest out of the vessel wall. TrackMate was also used to calculate the migratory path and the velocity of interstitial GFP[high] neutrophils. Data are from two independent experiments in which one mouse was used per condition.

## Western blotting

Purified neutrophils, treated as described in the main text, were washed once in PBS and lysed in ice-cold lysis buffer (50 mM NaCl, 50 mM Tris-HCl, 1% TritonX, 0.25% sodium deoxycholate supplemented with proteases and phosphatases inhibitor cocktails) for 30 min at 4°C. Solubilized and detergent-insolubilized proteins were separated by centrifugation at 14,000 × *g* for 15 min at 4°C. Then, 20 µg of total proteins (determined by Bicinchoninic acid assay, Thermo # 23225) were electrophoretically resolved on precast 10% polyacrylamide gels (Bio-Rad #4561034) and blotted onto a nitrocellulose membrane (Bio-Rad # 1704158) using a Trans-Blot Turbo Transfer System (Bio-Rad). Membranes were incubated ON at 4°C with the primary antibodies specified in the figure legends and revealed with HRP-conjugated secondary antibodies by the addition of ECL substrate (Clarity Western ECL substrate kit Bio-Rad #1705060S). The chemiluminescence signals were detected by X-ray film exposure (GE Healthcare Life Sciences #28906839).

## CD31[agonist] binding assay

Purified human or murine neutrophils were suspended at $10^7$/ml and treated with fMLP ($10^{-6}$ M) in suspension for 0, 2, 5, or 10 min at 37°C in the presence of a FITC-conjugated version of the CD31[agonist] peptide (50 µg/ml). At each time point, cells were washed to discard the unbound CD31[agonist] peptide and then immediately fixed with 2% PFA for 15 min at 4°C. After washing, cells were analyzed by flow cytometry using an LSR II flow cytometer (BD Biosciences). FITC MFIs were reported as a fold change from the unstimulated, baseline condition (time 0), and were used to evaluate the binding of the CD31[agonist] peptide to the neutrophil membrane. In other experiments, total internal reflection fluorescence microscopy (TIRF) was employed to evaluate the presence of CD31 molecular clusters and the distribution of the CD31[agonist] at the cellular membrane. Freshly isolated human neutrophils were stained with an A488-conjugated monoclonal antibody (clone MBC78.2) and with or without a rhodamine-conjugated form of the CD31[agonist] (50 µg/ml). Next, neutrophils were stimulated with fMLP ($10^{-6}$ M) or vehicle for 30 min, then washed and fixed with ice-cold 2% PFA at 4°C for 15 min. After fixation, cells were transferred to 35 mm high-grade glass-bottom poly-D-lysine-coated dishes (ibidi # 81158) and left to adhere for 2 hr in PBS. Digital images were acquired using the Zeiss TIRF 3 system using a αPlan-FLuAR ×100/1.45 Oil under the control of the AxioVision software. Laser lines 488 nm and 561 nm were used for CD31 and rhodamine-CD31[agonist] excitation, respectively. Three independent experiments with three different donors gave similar results.

## Statistical analysis

Each experiment encompassed a minimum of three individual conditions (cultured cells or mice) and was replicated at least twice, yielding consistent outcomes. The results are presented as mean ± SD or SEM, as indicated in the figure legend. The sample size was determined based on the guidelines set forth by local ethical committees and previous laboratory experiments, ensuring sufficient statistical power to detect the hypothesized effect. All statistical analyses were conducted using GraphPad

Prism Software (La Jolla, CA). Correlations were assessed using the Spearman correlation test and exponential regression. Group means were compared using the Mann–Whitney test, one-way and two-way ANOVA, or unpaired $t$-test, depending on the group sizes and variable distributions. Statistical significance was considered for $p < 0.05$ and denoted with asterisk symbols as follows: $*p < 0.05$, $**p < 0.01$, $***p < 0.001$, and $****p < 0.0001$.

## Acknowledgements

This work received support from multiple funding sources. We acknowledge the financial support from the following organizations: the Institut National de la Santé et de la Recherche Médicale (INSERM), the Université Paris Cité, a MSDAVENIR research grant (project 'Save-Brain'), the French National Research Agency (ANR) through the 'Investments for the future' program (projects 10-LABX-0017 'Inflamex' and DS0404-16-RHUS-00010 'iVASC'). Additionally, the work conducted in Glasgow was supported by the Engineering and Physical Sciences Research Council (EPSRC) grant EP/L014165/1 and the British Heart Foundation (BHF) grant PG/19/84/34771. We would like to acknowledge the Region Ile de France (CORDDIM) for providing doctoral grants to Drs. Andreata and Clement.

## Additional information

### Competing interests

Antonino Nicoletti: is an inventorn patents filed ("Improved CD31 peptides" PCT/EP2013/062806) related to the work presented in this manuscript. No other potential conflicts of interest relevant to this article exist. Giuseppina Caligiuri: is an inventor on patents filed ("Improved CD31 peptides" PCT/EP2013/062806) related to the work presented in this manuscript. No other potential conflicts of interest relevant to this article exist. The other authors declare that no competing interests exist.

### Funding

| Funder | Grant reference number | Author |
| --- | --- | --- |
| MSDAVENIR | SAVE-Brain | Antonino Nicoletti |
| Agence Nationale de la Recherche | 10-LABX-0017 "Inflamex" | Marc Clément |
| Agence Nationale de la Recherche | DS0404-16-RHUS-00010 "iVASC" | Giuseppina Caligiuri |
| Engineering and Physical Sciences Research Council | grant EP/L014165/1 | Pasquale Maffia |
| British Heart Foundation | PG/19/84/34771 | Pasquale Maffia |

The funders had no role in study design, data collection and interpretation, or the decision to submit the work for publication.

### Author contributions

Francesco Andreata, Juliette Hadchouel, Conceptualization, Resources, Data curation, Formal analysis, Supervision, Funding acquisition, Investigation, Methodology, Writing – original draft, Writing – review and editing; Marc Clément, Conceptualization, Data curation, Formal analysis, Investigation, Methodology, Writing – original draft, Writing – review and editing; Robert A Benson, Resources, Data curation, Investigation, Methodology, Writing – review and editing; Emanuele Procopio, Guillaume Even, Julie Vorbe, Véronique Ollivier, Investigation; Samira Benadda, Data curation; Benoit Ho-Tin-Noe, Conceptualization, Investigation; Marie Le Borgne, Data curation, Investigation, Writing – review and editing; Pasquale Maffia, Conceptualization, Resources, Data curation, Supervision, Methodology, Writing – review and editing; Antonino Nicoletti, Conceptualization, Resources, Data curation, Supervision, Funding acquisition, Investigation, Methodology, Writing – original draft, Project administration, Writing – review and editing; Giuseppina Caligiuri, Conceptualization, Resources, Data curation, Formal analysis, Supervision, Funding acquisition, Validation, Investigation, Visualization, Methodology, Writing – original draft, Project administration, Writing – review and editing

## Author ORCIDs

Marc Clément ⓘ https://orcid.org/0000-0002-6479-8360
Marie Le Borgne ⓘ https://orcid.org/0000-0002-3439-4867
Pasquale Maffia ⓘ https://orcid.org/0000-0003-3926-4225
Giuseppina Caligiuri ⓘ https://orcid.org/0000-0003-4973-2205

## Ethics

All the investigations were conformed to the directive 2010/63/EU of the European Parliament and formal approval was granted by the local Animal Ethics Committee (Comité d'étique Bichat-Debré, Paris, France).

## Decision letter and Author response

Decision letter https://doi.org/10.7554/eLife.84752.sa1
Author response https://doi.org/10.7554/eLife.84752.sa2

---

# Additional files

## Supplementary files

• Supplementary file 1. List of proteins identified by Orbitrap co-immunoprecipitate with CD31 in resting and fMLP-treated human neutrophils.

• MDAR checklist

## Data availability

All data generated or analysed during this study are included in the manuscript and supporting file. Source data files have been provided for Figures 1–5.

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
