## [Editor Report]

The demonstration of the hemophilic adhesion molecule CD31's localization to the migrating neutrophil's uropod and required signaling through its ITIM domain are valuable findings. Convincing evidence now includes defective extravasation of CD31 ITIM motif-deleted neutrophils in adoptive transfer experiments. Based on the critical role of leukocyte transmigration in many physiologic and pathologic processes, this work will be of interest to a broad audience.

---

## [Decision Letter]

**Decision letter after peer review:**

Thank you for submitting your article "CD31 signaling promotes the detachment at the uropod of extravasating neutrophils allowing their migration to sites of inflammation" for consideration by *eLife*. Your article has been reviewed by 2 peer reviewers, and the evaluation has been overseen by a Reviewing Editor and Carla Rothlin as the Senior Editor. The following individual involved in the review of your submission has agreed to reveal their identity: Jacopo Di Russo (Reviewer #1).

Thank you for submitting your manuscript to *eLife*. The topic of your work, that of molecular dynamics at the trailing edge (uropod) of migrating/extravasating neutrophils, is highly significant. Both reviewers found your paper of interest and importance. Major concerns include the correlative nature of the CD31 phosphorylation data. That is, many molecular signaling events occur when leukocytes vectorially migrate in response to extracellular cues, typically present during extravasation. They include Rho/Rac activation, actin oscillations, and actomyosin activation with force-generated propulsion of the MTOC toward the cell trailing edge. Within the hierarchy of all these events, it is not clear how relatively (and directly) important the noted CD31 tyrosine phosphorylation is in the forward migratory movement. Another major issue, raised by both reviewers, is that the in vivo CD31 KO data interpretation is complicated by the presence of CD31 on both myeloid and endothelial cells. Of the 2 options recommended, adoptive transfer of CD31-/- neutrophils into a WT mouse should be more easily and readily achievable than generating another genetic strain (neutrophil-specific CD31 KO). Reviewer 1's question about the interpretation of the RGD blocking peptide experiments is valid, since ICAM-1-binding beta2 integrins do not utilize the RGD sequence for that adhesion, making the compensation of uropod detachment in CD31-/- neutrophils difficult to understand. If not explained better, those experiments only confound the other data. Despite the cause-effect caveat/concern, we are interested in this work and would consider a revised version of the manuscript, addressing the Reviewers' concerns. A majority of those concerns relate to data quantification and presentation, so as to enhance the clarity and significance of the results. A few additional experiments would be necessary.

Essential revisions:

1) Address all Reviewer data clarification and qualification recommendations.

2) Dissect neutrophil vs. endothelial CD31 importance within the in vivo experiments – adoptive transfer of KO neutrophils would be an acceptable approach.

3) Much more thorough explanation of the inhibitory RGD peptide compensation experiments. If a good explanation cannot be presented, those experiments should be removed.

*Reviewer #1 (Recommendations for the authors):*

Andreata F. and colleagues in this work address the function of CD31 (PECAM-1) intracellular signalling in neutrophils in regulating integrins adhesion during extravasation. To explore the hypothesis of a key role of CD31 for cell rear (uropod) detachment in the final step of extravasation, they use a combination of mouse genetics, intravital microscopy, and in vitro cell migration assay. Convincing evidence shows that the phosphorylation of CD31 intracellular motive "ITIM" at the neutrophil's uropod supports its detachment both in vivo and in vitro. Nevertheless, the data showing a direct effect of the investigated pathway in reducing integrin-biding affinity to endothelial ICAM-1 or basement membrane laminins remains mostly correlative. In conclusion, this work, for the first time, shows the role of CD31 phosphorylation for uropod detachment and represents an important step forward in dissecting the different mechanisms regulating the complexity of the leukocyte extravasation process.

– The correlation between neutrophils position and phosphorylation levels of CD31 in Figure 1E looks convincing. Nevertheless, the quantification of the data using a linear correlation with the Pearson coefficient is not convincing due to their distribution. I would rather use an exponential correlation.

– The authors state that CD31 -/- neutrophils fail to migrate further from the extravasating vessel and remain close to it. This should not only be shown with a representative image but also with quantification, e.g., the percentage of total cells in close proximity to a vessel.

– The authors show the re-localization of CD31 and CD11b (Integrin α M) at the uropod during neutrophil migration (Figure 2F). This does not necessarily indicate that integrin β 2 – containing heterodimers, including CD11a/CD18 (Integrin α L β 2), localise there and mediated neutrophils adhesion. To strengthen the paper's conclusion, it would be necessary to show that integrin β 2 mostly relocates there and that active integrin β 2 is mostly present at the front. Furthermore, these conclusions would need all quantitative evidence, i.e., also for figure 2F. Please note that integrin α M and integrin β 2 have been shown to function for cell adhesion independently of their dimerization. https://doi.org/10.1074/jbc.M406968200.

– The data presented and plotted in Figures S3C and E should be tested with one and 2-way ANOVA, respectively, and not a t-test.

– The statement that GFP+ leukocytes have more elongated uropod and delay their detachment in the presence of CD31 agonist needs to be supported not only by videos but also by quantitative data. Furthermore, it no clear neither from the text, figure caption, or method how many independent experiments were performed for this experiment.

– In the last chapter of the manuscript, the authors suggest that CD31 activation is involved in a general reduction of the integrins activation state. This is shown by the injection of RGD peptide to antagonise integrin-ECM adhesion during CD31-/- neutrophils migration. I do not doubt the validity of this very interesting finding, but I struggle to understand the direct connection with the logic of the manuscript. In the previous chapter, the authors address the role of CD31 in regulating adhesion on integrin β 2 – ICAM-1 (Figure 3) and unspecified integrins on laminin α 4 (Figure 4). Integrin α L β 2, α M β 2 and laminin α 4-binding integrins (e.g., integrin α 6 β 1) do not bind their ligands through an RGD sequence. I would suggest directly addressing integrin β 2-ICAM-1 and integrin α 6 β 1-laminin α 4 interaction in vivo (or in vitro) using blocking antibodies or peptides. On the other hand, the generalization of the CD31-mediated uropod detachment via the deactivation of ECM-integrin adhesion is very interesting. If not with some proof-of-concept experiments, the crosstalk of CD31 activation and RGD-binding integrins (e.g., integrin α 5 β 1) should be addressed in the discussion.

*Reviewer #2 (Recommendations for the authors):*

This study investigates the role of CD31 in neutrophil transmigration. The authors use a CD31 knockout mouse to confirm that this receptor is required for neutrophil extravasation and then go on to use proteomics to identify the CD31 interactome, which reveals integrins and multiple regulators of the actin cytoskeleton. Using microscopy, the authors compellingly show that, in neutrophils, CD31 localizes to the uropod and where it promotes termination of integrin signaling (required for uropod detachment). The manuscript is clear and well-written and the conclusions are generally robust. This is an important mechanistic addition to our understanding of neutrophil extravasation. in vivo conclusions, however, are complicated by the fact that CD31 is abundant in endothelial cells so it becomes difficult to disentangle the contribution of the neutrophil versus the endothelial CD31 pool.

Figure 1: It is difficult to conclude a neutrophil-intrinsic defect here when endothelial CD31 affects extravasation. The authors should test a neutrophil-specific knockout, or (more easily) adoptively transfer into WT mice.

Figure 1F: the authors write that CD31-/- neutrophils " failed to further migrate since they remained in close contact to the outer layer of the vessel wall". This is hard to conclude from the single image and should be quantified. Also, it appears from the image that there are fewer neutrophils in the vessel, which would argue for a recruitment defect. Quantitative data would greatly clarify the phenotype in this analysis.

Figure 1C: what is the rationale for looking at those cytokines? It would be better to include the ones that are linked to IL-1, such as IL-6 and GCSF.

Figure 1F: the authors write that CD31-/- neutrophils " failed to further migrate since they remained in close contact to the outer layer of the vessel wall". This is hard to conclude from the single image and should be quantified. Also, it appears from the image that there are fewer neutrophils in the vessel, which would argue for a recruitment defect. Quantitative data would greatly clarify the phenotype in this analysis.

Figure 4A-D: the most striking effect of the agonist is that there appear to be more neutrophils in the vessel. I recommend that this is quantified and that you measure circulating neutrophil counts as well as GCSF levels, otherwise it is difficult to make conclusions about extravasation.

Figure 4E-F: how do cell length and circularity correspond to migration capacity? This is not explained.

How do authors exclude macrophages in lysM-gfp mice?

Figure 5: how do you exclude the effect of endothelial cd31? Adoptive transfer experiments are needed.

---

## [Author Response]

Essential revisions:Reviewer #1 (Recommendations for the authors):– The correlation between neutrophils position and phosphorylation levels of CD31 in Figure 1E looks convincing. Nevertheless, the quantification of the data using a linear correlation with the Pearson coefficient is not convincing due to their distribution. I would rather use an exponential correlation.

We appreciate the constructive comment from the reviewer, highlighting a potential lack of clarity in the methods description and data visualization related to Figure 1E. To address this concern, we conducted a nonparametric Spearman correlation analysis and utilized an exponential regression model to establish the correlation between pY713 and the distance of neutrophils from the vessel. Furthermore, we have included a new Figure 1E, along with an updated figure legend and methods that accurately reflects these changes.

– The authors state that CD31 -/- neutrophils fail to migrate further from the extravasating vessel and remain close to it. This should not only be shown with a representative image but also with quantification, e.g., the percentage of total cells in close proximity to a vessel.

In response to the reviewer's suggestions, we have now measured both the distance of cells from the vessels and the percentage of cells in close proximity to them in both *Pecam1^-/-^* and WT mice. These additional measurements allowed us to generate two new graphs, namely Figure 1F and Figure 1H. The inclusion of these quantifications has further supported our claim that *Pecam1^-/-^* neutrophils tend to remain in close proximity to the extravasating vessel and exhibit limited migration away from it. Consequently, we have updated the figure legend to accurately reflect these changes.

– The authors show the re-localization of CD31 and CD11b (Integrin α M) at the uropod during neutrophil migration (Figure 2F). This does not necessarily indicate that integrin β 2 – containing heterodimers, including CD11a/CD18 (Integrin α L β 2), localise there and mediated neutrophils adhesion.

We appreciate the reviewer's comment regarding the re-localization of CD31 and CD11b (Integrin α M) at the uropod during neutrophil migration as shown in Figure 2F. It is important to note that integrins are heterodimers composed of α and β subunits. In the case of CD11b, it forms a complex with the β2 subunit, CD18, to create the functional integrin αMβ2 (also known as Mac-1). Without its β subunit partner, CD11b is unlikely to be stably expressed on the cellular membrane. Studies have demonstrated that integrins require their β subunit for proper folding, trafficking, and function (Kim M, Carman CV, Springer TA, 2003*)*. Therefore, it is unlikely that CD11b would be expressed alone on the cellular membrane without its β subunit partner. We apologize for any confusion caused and believe that this clarification is addressed in the revised manuscript (page 8).

To strengthen the paper's conclusion, it would be necessary to show that integrin β 2 mostly relocates there and that active integrin β 2 is mostly present at the front. Furthermore, these conclusions would need all quantitative evidence, i.e., also for figure 2F. Please note that integrin α M and integrin β 2 have been shown to function for cell adhesion independently of their dimerization. https://doi.org/10.1074/jbc.M406968200.

We would like to emphasize that our study provides evidence supporting the relocalization of β2 subunits involved in the active state of integrins during neutrophil migration, as demonstrated in Figure 2J. This figure includes data illustrating the localization of active LFA-1, a heterodimer formed by the α L and β2 integrin subunits, at the leading edge of neutrophils.

To assess the active conformation of the β2 integrin, we employed a conformation-specific antibody (Clone NKI-L16) that specifically recognizes an epitope of LFA-1 in its extended, active conformation. Consequently, our findings consistently indicate that the active form of β2 integrins, exemplified by active LFA-1, is predominantly present at the front of migrating neutrophils, thereby providing support for our conclusions.

In response to the mentioned reference (https://doi.org/10.1074/jbc.M406968200) discussing the potential independent functions of integrin α M (CD11b) and integrin β 2, we acknowledge its relevance. However, it is important to clarify that our study primarily focuses on investigating the role of CD31 in modulating neutrophil migration and the dynamics of integrin molecular complexes.

In our research, we highlight the relocation of CD31 at the uropod, where CD11a β2 integrins assume a closed, inactive conformation. This observation emphasizes the significance of CD31 in influencing the activity and distribution of integrin complexes during neutrophil migration.

We sincerely appreciate the reviewer's attention to this aspect of our study, and we concur that the additional details provided in the Results section on *page 8* and the legend to Figure 2 further bolster our conclusions.

– The data presented and plotted in Figures S3C and E should be tested with one and 2-way ANOVA, respectively, and not a t-test.

Thank you for your suggestion. We have performed the suggested one-way and two-way ANOVA tests on the data presented in Figures S3C and E. Consequently, we have updated the legend of the revised Figure 4—figure supplement 1 to reflect these appropriate statistical analyses. We appreciate your valuable input in improving the statistical analysis of our results.

– The statement that GFP+ leukocytes have more elongated uropod and delay their detachment in the presence of CD31 agonist needs to be supported not only by videos but also by quantitative data. Furthermore, it no clear neither from the text, figure caption, or method how many independent experiments were performed for this experiment.

We appreciate the reviewer's comment and thank them for their suggestion. To address this concern, we have taken the reviewer's advice into account. We have now quantified the velocity of individual neutrophils after they have detached from the vessel wall, similar to how it was done for the 3-hour timepoint. In order to ensure accuracy, we utilized a region of interest during our analysis, focusing exclusively on neutrophils detaching from the vessel. This approach has allowed us to provide quantitative data that supports the observation that the CD31 agonist enhances detachment speed. To reflect these new findings, we have created a new graph, Figure 4C, and updated the figure legend accordingly. Additionally, in the method section, we have clarified the number of independent experiments conducted.

– In the last chapter of the manuscript, the authors suggest that CD31 activation is involved in a general reduction of the integrins activation state. This is shown by the injection of RGD peptide to antagonise integrin-ECM adhesion during CD31-/- neutrophils migration. I do not doubt the validity of this very interesting finding, but I struggle to understand the direct connection with the logic of the manuscript. In the previous chapter, the authors address the role of CD31 in regulating adhesion on integrin β 2 – ICAM-1 (Figure 3) and unspecified integrins on laminin α 4 (Figure 4). Integrin α L β 2, α M β 2 and laminin α 4-binding integrins (e.g., integrin α 6 β 1) do not bind their ligands through an RGD sequence. I would suggest directly addressing integrin β 2-ICAM-1 and integrin α 6 β 1-laminin α 4 interaction in vivo (or in vitro) using blocking antibodies or peptides. On the other hand, the generalization of the CD31-mediated uropod detachment via the deactivation of ECM-integrin adhesion is very interesting. If not with some proof-of-concept experiments, the crosstalk of CD31 activation and RGD-binding integrins (e.g., integrin α 5 β 1) should be addressed in the discussion.

After carefully considering the reviewer's feedback, we have revised our manuscript to address the concerns raised. We agree that directly assessing the impact of CD31 signaling on integrin β 2-ICAM-1 and integrin α 6 β 1-laminin α 4 binding, which are not mediated by the RGD motif, is crucial for evaluating its global role. We acknowledge that our data only support direct evidence for a role of RGD-binding integrins in the impaired migration of Pecam1-/- neutrophils. Unfortunately, due to time constraints, we were unable to conduct the suggested experiments within the scope of this study.

Nevertheless, we recognize the importance of this point and have taken steps to address this limitation in the revised manuscript. On page 13-15 of the extensively revised discussion, we highlight the need for further studies specifically investigating the interactions between CD31 activation and integrin β 2-ICAM-1, as well as integrin α 6 β 1-laminin α 4. We provide a more detailed explanation of our interpretation regarding the broader activity of soluble RDG peptides in our rescue experiments, as suggested by the reviewer.

We sincerely appreciate the insightful comments from the reviewer, and although we couldn't include the additional experiments, we believe that our revised manuscript offers a more comprehensive discussion of the implications and potential avenues for future research.

Reviewer #2 (Recommendations for the authors):Figure 1: It is difficult to conclude a neutrophil-intrinsic defect here when endothelial CD31 affects extravasation. The authors should test a neutrophil-specific knockout, or (more easily) adoptively transfer into WT mice.

We appreciate the reviewer for raising this important point. In a previous study by Dangerfield et al. 2003, the relative contribution of endothelial vs. neutrophil CD31 to the extravasation of the latter was precisely investigated. In their work (Figure 7a), bone marrow chimeras were used to generate CD31 deficiency either in the endothelial cells or in the hematopoietic system. Interestingly, the migratory defect was not rescued in either case, indicating that CD31 trans-homophilic interaction between endothelial cells and neutrophils is necessary for proper neutrophil migration.

However, prompted by the reviewer's suggestion, we conducted adoptive transfer experiments to investigate another unanswered question: whether CD31 intracellular signaling, rather than the mere presence of the protein, regulates neutrophil recruitment to inflammatory sites. To address this question, we conducted a new experiment in which we adoptively transferred WT or our newly-generated *Pecam1*^ITIM-/-^ neutrophils in a 1:1 ratio into WT recipient mice. Before injection, we performed differential CFSE/CTV staining to distinguish the origin of the neutrophils. After 24 hours, IL-1β was intraperitoneally administered to the recipient mice, and the transmigrated neutrophils were collected from the peritoneal wash 4 hours later in this competition assay (new Figure 5I). Despite equal frequencies of WT and *Pecam1*^ITIM-/-^ neutrophils (identified as CD45^+^ Ly6G^+^ by flow cytometry) in the blood before and after IL-1β injection (new Figure 5J-K), *Pecam1*^ITIM-/-^ neutrophils exhibited a disadvantage in recruitment compared to their WT counterparts (new Figure 5L). These findings provide novel evidence that the CD31 ITIM-dependent intracellular pathway plays an important role in controlling neutrophil migration.

We believe that these additional results significantly contribute to our understanding of CD31-mediated neutrophil recruitment and highlight the importance of CD31 intracellular signaling in this process.

Figure 1F: the authors write that CD31-/- neutrophils " failed to further migrate since they remained in close contact to the outer layer of the vessel wall". This is hard to conclude from the single image and should be quantified. Also, it appears from the image that there are fewer neutrophils in the vessel, which would argue for a recruitment defect. Quantitative data would greatly clarify the phenotype in this analysis.

Based on the reviewer's suggestions, we have conducted additional measurements to address their concerns. Specifically, we quantified the distance of neutrophils from the vessels and determined the percentage of cells located in close proximity to the vessel wall in both *Pecam1*^-/-^ and WT mice. These quantitative data have been incorporated into two new graphs, namely Figure 1F and Figure 1H. As a result, the statement that "*Pecam1*^-/-^ neutrophils failed to further migrate since they remained in close contact to the outer layer of the vessel wall" is now supported by robust quantification. To reflect these updates, we have revised the figure legend accordingly. We thank the reviewer for this suggestion, as it prompted us to provide quantitative data that enhance the clarity and strength of our conclusions regarding the migratory behavior of *Pecam1*^-/-^ neutrophils.

Figure 1C: what is the rationale for looking at those cytokines? It would be better to include the ones that are linked to IL-1, such as IL-6 and GCSF.

We appreciate the reviewer's feedback regarding the selection of cytokines in Figure 1C, and we acknowledge the importance cytokines specifically linked to IL-1, such as IL-6 and GCSF. However, the intention behind our chosen markers was to assess potential specific mechanisms underlying the different recruitment of neutrophils observed between WT and CD31 mice in response to IL-1β. In this perspective, PTX-3 was chosen to assess whether the difference relied on a different liver acute phase response to circulating IL-1β (Skelly DT, et al., 2013). SDF-1 was intended to explore a potential different triggering of neutrophil mobilization from the bone marrow (Suratt BT, et al., 2004), and CXCL1 was meant to assess the potential different extent of chemoattraction specific to neutrophils (Sawant KV, et al., 2016). CD62P was chosen as its release as a soluble marker reflects the extent of local platelet/endothelial activation, which is a prerequisite for blood neutrophil rolling and migration at specific vascular sites (Zuchtriegel G, et al., 2015).

We have incorporated this rationale into the results paragraph referring to Figure 1C in the revised manuscript (page 6).

Figure 1F: the authors write that CD31-/- neutrophils " failed to further migrate since they remained in close contact to the outer layer of the vessel wall". This is hard to conclude from the single image and should be quantified. Also, it appears from the image that there are fewer neutrophils in the vessel, which would argue for a recruitment defect. Quantitative data would greatly clarify the phenotype in this analysis.Figure 4A-D: the most striking effect of the agonist is that there appear to be more neutrophils in the vessel. I recommend that this is quantified and that you measure circulating neutrophil counts as well as GCSF levels, otherwise it is difficult to make conclusions about extravasation.

Please note that the inflammatory stimulus in these experiments was localized to the ear, and the snapshots shown in Figure 4A-D are derived from multiphoton time-lapse imaging. The purpose of this analysis was to evaluate the local trafficking of neutrophils. The agonist consistently increases the speed of neutrophil flow in the analyzed vessel sections. As a result, the higher density of neutrophils captured in the images with the agonist reflects the free-flowing state of most neutrophils within the vessel lumen, while stably adhering neutrophils appear to be fewer in number because they are marginated.

To directly address the reviewer's concern, we performed a repeat experiment in which we administered IL-1β directly into the ears of C57Bl6 mice. Unfortunately, the *Lyz2*-GFP^+^ mice that were previously used in our study are no longer available in our laboratory. We counted the number of neutrophils in fresh heparinized blood using the ABC VET automatic formula and measured several soluble plasma parameters, including IL-6 and GM-CSF, using Thermofisher Mouse ProcartaPlex Simplex Kits. Untouched mice (no challenge) served as control (3 groups, n=4/group). These additional measurements allowed us to assess the systemic effects of the CD31 agonist, as requested by the reviewer. These experiments revealed that the neutrophil count (% of white blood cells) was not significantly influenced by either the local inflammatory process or the CD31 agonist (One-way ANOVA, p = 0.3856). Additionally, the soluble parameters, including IL-6 and GM-CSF, were not detectable in the systemic circulation across all three groups. For the reviewer's inspection, the additional data are presented in Author response image 1.

**Author response image 1. sa2fig1:** 

Figure 4E-F: how do cell length and circularity correspond to migration capacity? This is not explained.

We apologize to the reviewer for the lack of clarity in explaining the relationship between cell length and circularity and migration capacity. Longer cells have a higher surface area to volume ratio, which can facilitate increased interaction with the extracellular matrix and potentially enhance adhesion. This characteristic may be particularly important for cells that need to migrate through narrow spaces, such as cancer cells invading tissue barriers or neutrophils navigating extracellular surfaces. By measuring cell length and circularity, we aimed to gain insights into the morphological features that could influence migration capacity. We understand that further explanation was necessary and have provided a more detailed discussion in the revised manuscript (Results, page 10-11).

How do authors exclude macrophages in lysM-gfp mice?

We appreciate the reviewer's inquiry regarding the exclusion of macrophages in our study using *Lyz2*-GFP mice. In this model, macrophages can be distinguished from granulocytes based on the difference in GFP fluorescence intensity. Previous research has shown that the GFP fluorescence intensity of macrophages is significantly lower compared to that of granulocytes (Faust N, et al., 2000). By carefully examining the GFP intensity, we were able to exclude macrophages from our analysis. This information has been added into the corresponding method section.

Figure 5: how do you exclude the effect of endothelial cd31? Adoptive transfer experiments are needed.

We appreciate the reviewer for raising this important point. In a previous study by Dangerfield et al., 2002, the relative contribution of endothelial vs. neutrophil CD31 to the extravasation of the latter was precisely investigated. In their work (Figure 7a), bone marrow chimeras were used to generate CD31 deficiency either in the endothelial cells or in the hematopoietic system. Interestingly, the migratory defect was not rescued in either case, indicating that CD31 trans-homophilic interaction between endothelial cells and neutrophils is necessary for proper neutrophil migration.

However, prompted by the reviewer's suggestion, we conducted adoptive transfer experiments to investigate another unanswered question: whether CD31 intracellular signaling, rather than the mere presence of the protein, regulates neutrophil recruitment to inflammatory sites. To address this question, we conducted a new experiment in which we adoptively transferred WT or our newly-generated *Pecam1*^ITIM-/-^ neutrophils in a 1:1 ratio into WT recipient mice. Before injection, we performed differential CFSE/CTV staining to distinguish the origin of the neutrophils. After 24 hours, IL-1β was intraperitoneally administered to the recipient mice, and the transmigrated neutrophils were collected from the peritoneal wash 4 hours later in this competition assay (new Figure 5I). Despite equal frequencies of WT and *Pecam1*^ITIM-/-^ neutrophils (identified as CD45^+^ Ly6G^+^ by flow cytometry) in the blood before and after IL-1β injection (new Figure 5J-K), *Pecam1*^ITIM-/-^ neutrophils exhibited a disadvantage in recruitment compared to their WT counterparts (new Figure 5L). These findings provide novel evidence that the CD31 ITIM-dependent intracellular pathway plays an important role in controlling neutrophil migration.

We believe that these additional results significantly contribute to our understanding of CD31-mediated neutrophil recruitment and highlight the importance of CD31 intracellular signaling in this process.